# What chemical species are responsible for new particle formation and growth in the Netherlands? A hybrid positive matrix factorization (PMF) analysis using aerosol composition (ACSM) and size (SMPS)

Farhan R. Nursanto[1], Roy Meinen[2], Rupert Holzinger[2], Maarten C. Krol[1,2], Xinya Liu[3], Ulrike Dusek[3], Bas Henzing[4], Juliane L. Fry[1]

[1]Meteorology and Air Quality (MAQ), Environmental Sciences Group, Wageningen University and Research (WUR), Wageningen, 6708PB, the Netherlands
[2]Institute for Marine and Atmospheric Research Utrecht, Department of Physics, Utrecht University, Princetonplein 5, 3584CC, Utrecht, the Netherlands
[3]Centre for Isotope Research (CIO), Energy and Sustainability Research Institute Groningen (ESRIG), University of Groningen, Groningen 9747 AG, the Netherlands
[4]Netherlands Organisation for Applied Scientific Research (TNO), Princetonlaan 6, 3584 Utrecht, the Netherlands

*Correspondence to*: Juliane L. Fry (juliane.fry@wur.nl)

**Abstract.** Aerosol formation acts as a sink for gas-phase atmospheric species that controls their atmospheric lifetime and environmental effects. To investigate aerosol formation and evolution in the Netherlands, a hybrid positive matrix factorization (PMF) analysis has been conducted using observations from May, June, and September 2021 collected in a rural site of Cabauw in Central Netherlands. The hybrid input matrix consists of the full organic mass spectrum acquired from a time-of-flight aerosol chemical speciation monitor (ToF-ACSM), ACSM inorganic species concentrations, and binned particle size distribution concentrations from a scanning mobility particle sizer (SMPS). These hybrid PMF analyses discerned four factors that describe aerosol composition variations: two size-driven factors that are related to new particle formation (NPF) and growth (F4 and F3), and two bulk factors driven by composition, not size (F2 and F1). The distribution of chemical species across these factors shows different compounds responsible for nucleation and growth of new particles. The smallest-diameter size factor (F4) contains ammonium sulfate and organics, and typically peaks during the daytime. Newly formed particles, represented by F4, are correlated mainly with wind from the southwesterly-westerly and easterly sectors that transport sulfur oxides ($SO_x$), ammonia ($NH_3$), and organic precursors to Cabauw. As the particles grow from F4 to F3 and bulk factors, nitrate and organics plays an increasing role, and the particle loading diurnal cycle shifts from daytime to a nighttime maximum. Greater organics availability makes secondary organic aerosol (SOA) more influential in summertime aerosol growth, principally due to volatility differences produced by seasonal variation in photooxidation and temperature.

Keywords: new particle formation (NPF), positive matrix factorization (PMF), particle size distribution, sulfate aerosol, nitrate aerosol, organic aerosol

## 1. Introduction

Atmospheric aerosols are solid or liquid particles suspended in the air that are formed from natural or anthropogenic sources (Haywood, 2016). To describe the aerosol particle size distribution, four modes are generally distinguished according to the particle geometric diameter ($D_p$): nucleation mode ($D_p < 20$ nm), Aitken mode (20 nm $< D_p <$ 100 nm), accumulation mode (90 nm $< D_p <$ 1000 nm), and coarse mode ($D_p > 1$ μm) (Hussein et al., 2004; Wu et al., 2008). New particle formation (NPF) is identified by a rapid buildup of high atmospheric concentrations of aerosol particles in the nucleation mode. These particles subsequently grow into Aitken mode particles and further (Maso et al., 2005; Spracklen et al., 2010; Salimi et al., 2015; Kerminen et al., 2018; Lee et al., 2019).

Aerosols impact the Earth by absorbing and scattering solar and terrestrial radiation (Andreae and Crutzen, 1997; Grantz et al., 2003; Wong et al., 2017; Marrero-Ortiz et al., 2019), and indirectly by producing or modifying clouds (Lohmann and Feichter, 2005; Mahowald et al., 2011; Fan et al., 2018). NPF plays a prominent role in cloud formation by contributing to over 50% of cloud condensation nuclei formation, which affects the lifetime and radiative properties of clouds (Bianchi et al., 2016; Gordon et al., 2016; Haywood, 2016; Dall'Osto et al., 2018; Lee et al., 2019). These phenomena affect the ecosystem physically by modifying radiation diffusion, temperature, and precipitation (Grantz et al., 2003; Haywood, 2016; Lee et al., 2019). Aerosols also influence the ecosystem chemically through influencing the spatial patterns of nitrogen deposition (van der Swaluw et al., 2011; Wamelink et al., 2013) and oxidative processes (Xing et al., 2017), leading to ecological harm such as soil pollution, water acidification, eutrophication, and loss of biodiversity (Erisman et al., 2011; Wamelink et al., 2013). In terms of public health, aerosols exhibit adverse effects on human health due to their size and chemical composition. NPF events are typically followed by air quality degradation, which is consistently associated with elevated pulmonary and cardiovascular morbidity and mortality worldwide (Ayala et al., 2012; Pope et al., 2020).

Sulfuric acid ($H_2SO_4$) is typically understood to be the most prevalent nucleation-inducing agent in NPF events, together with other airborne chemical species, including nitric acid ($HNO_3$), bases (e.g., amines), and organic acids (Zhang et al., 2012; Kulmala et al., 2013; Zhang et al., 2015; Wagner et al., 2017; Lehtipalo et al., 2018; Kürten, 2019; Lee et al., 2019; Brean et al., 2021; Olin et al., 2022). Numerous studies also report low-volatility organic species, such as terpene oxidation products and organic nitrates, participating in the formation of new particles (Berkemeier et al., 2016; Bianchi et al., 2016; Tröstl et al., 2016; Barsanti et al., 2017; Dall'Osto et al., 2018; Kerminen et al., 2018; Lee et al., 2019; Heinritzi et al., 2020).

In this work, we show that co-located measurements of aerosols' atmospheric composition and particle size distribution can be used to characterize the chemical composition of new particle and aerosol components that facilitate their growth. A time-of-flight aerosol chemical speciation monitor (ToF-ACSM, Aerodyne Inc.) allows the continuous and real-time quantification of non-refractory chemical species in ambient air (Ng et al., 2011; Fröhlich et al., 2013). For particle size distributions, the

scanning mobility particle sizer (SMPS) provides real-time measurement of submicron particle number concentrations of different sizes (Amaral et al., 2015; Wiedensohler et al., 2012).

Aerosol mass spectrometry measurements have been used extensively with positive matrix factorization (PMF) as a strategy for aerosol source apportionment, especially regarding the organic components (Lanz et al., 2007; Jimenez et al., 2009; Ulbrich

et al., 2009; Ng et al., 2010; Zhang et al., 2011). This paper combines the organic mass spectrum and chemical species concentrations from ToF-ACSM with particle size distribution from SMPS into a hybrid PMF input matrix, to study the association between chemical composition and particle size distribution. A similar approach for hybrid ACSM-SMPS PMF analysis was used for a European aerosol dataset comparison (Dall'Osto et al., 2018). Previous studies on aerosol source apportionment in the Netherlands have focused on organic aerosol composition (Mooibroek et al., 2011; Mensah et al., 2012;

Schlag et al., 2016). Here, we analyze ACSM-SMPS datasets from Cabauw, the Netherlands, collected as part of the Ruisdael Observatory Land-Atmosphere Interactions Intensive Trace-gas and Aerosol (RITA) campaign in May to September 2021 (https://ruisdael-observatory.nl), using PMF to characterize the chemical species responsible for new particle formation and growth across several seasons. Several studies have shown NPF events dependent on air mass origin transporting different pollutants (Hamed et al., 2007; Modini et al., 2009; Castro et al., 2010; Asmi et al., 2011; Németh and Salma, 2014; Nieminen

et al., 2014; Qi et al., 2015; Mordas et al., 2016; Kolesar et al., 2017; Peng et al., 2017; Kerminen et al., 2018; Pushpawela et al., 2019), and therefore we also explore relationships between wind direction, wind speed, and factor time series to interpret source apportionment.

## 2. Methods and Instrumentation

### 2.1. Cabauw site and meteorological conditions

Measurements were performed at the CESAR tower (51.970° N, 4.926° E), managed and operated by the Royal Netherlands Meteorological Institute (KNMI, the Netherlands) (see Fig. 1a). The tower is located near Cabauw, in the province of Utrecht, the Netherlands, approximately 18 km southwest of Utrecht city center, 31 km east of the city and (largest in Europe) port of Rotterdam, 45 km south of Amsterdam, and 45 km southeast of the Dutch North Sea coast. To the east and south of the site are the provinces of Gelderland and Noord-Brabant, which consist mainly of forests, agricultural lands with clay and sand soil

types for crops, and animal farms, specifically chicken and pig farms in the south and cattle in the east (CBS, 2022). The Cabauw site itself is rural and surrounded by agricultural lands. The dataset used in this analysis contains overlapping ACSM and SMPS data split into periods from 11–31 May 2021, 1–22 June 2021, and 1–30 September 2021, providing some seasonal variation. To simplify, we will refer to these periods as May, June, and September, respectively, in this paper.

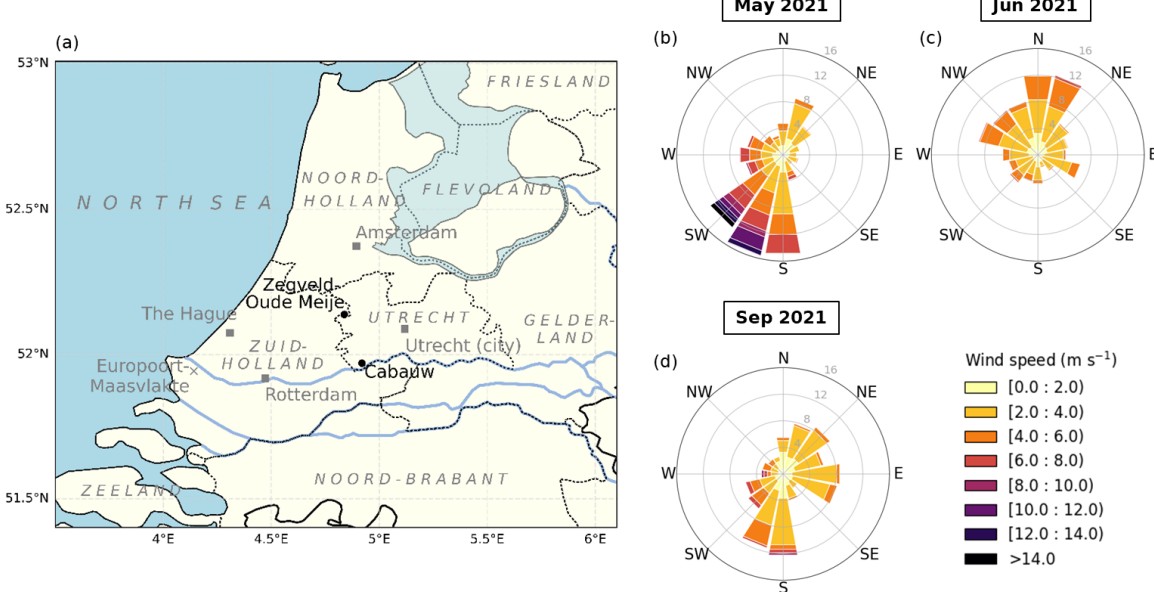


**Figure 1. (a)** Map of a part of the Netherlands showing the locations of the measurement stations Cabauw (main site) and Zegveld-Oude Meije (auxiliary NH₃ measurements). The province, sea, and neighboring country names are indicated in italic and light grey. The big cities of Amsterdam, The Hague, Rotterdam, and Utrecht in the area collectively known as "Randstad" are situated in Noord-Holland, Zuid-Holland, and Utrecht provinces. The urban and harbor area of Rotterdam extends as Europoort-Maasvlakte to the mouth of the Maas River.

**(b-d)** Wind rose plots for May, June, and September 2021. Winds from S up to SW sector were dominant in May. In June, the prevailing winds were from WNW up to the NNE sector. In September, two major wind directions were from the E and S sectors.

Weather data were retrieved from the Royal Netherlands Meteorological Institute (KNMI, https://www.knmi.nl). In general, May 2021 was characterized by moderate spring temperatures (11.8 °C on average) with scattered precipitation transitioning

into the warmer summer period. June 2021 had the highest temperatures (18.7 °C on average) and was the sunniest of the three periods, reflecting summer weather. September 2021 showed warm temperatures (16.2 °C on average), with less radiation and precipitation compared to May 2021. Winds from south to southwest (180° to 225°) dominated in spring (May), bringing plumes from the agriculture-heavy province of Noord-Brabant. In summer (June), the prevailing air masses were coming from west-northwest to north-northeast (292.5° to 22.5°), bringing air from the North Sea and some major cities along the coast

and/or in the Randstad, such as Rotterdam, The Hague, Amsterdam, and Utrecht. More diverse wind plumes were observed in September, ranging from easterly (22.5° to 112.5°), coming from the forested nature and agricultural areas in the province of Gelderland, and southerly (180° to 202.5°), coming from the province of Noord-Brabant. The meteorological variables for each period are summarized in Table 1 and the wind variables are visualized as wind roses in Fig. 1b-d.


**Table 1.** Meteorological conditions on three periods analyzed (May, June, and September 2021). The periods represent spring, summer, and autumn weather, respectively.

| Period | Temperature (ºC) | Downward shortwave radiation (W m$^{-2}$) | Precipitation (mm) |
|---|---|---|---|
| 11–31 May 2021 | 11.8 (mean) <br> 3.5 (low), 23.0 (high) | 211.7 (mean) <br> 1032.2 (max) | 96.6 (total) |
| 1–22 Jun 2021 | 18.7 (mean) <br> 8.1 (low), 29.6 (high) | 264.9 (mean) <br> 957.2 (max) | 31.2 (total) |
| 1–30 Sep 2021 | 16.2 (mean) <br> 5.7 (low), 26.7 (high) | 142.0 (mean) <br> 818.9 (max) | 24.4 (total) |

## 2.2. Measurement setup

### 2.2.1. Chemical species measurements

The ToF-ACSM was the main instrument employed, allowing the analysis of non-refractory organics, ammonium, nitrate, sulfate, and chloride in the aerosol phase. The instrument has been detailed in other work (Fröhlich et al., 2013). Ambient air was drawn into the instrument through a stainless-steel tubing inlet system equipped with a PM$_{2.5}$ size-cut cyclone (URG-2000-30ED) and a Nafion dryer, sampling at 4.5-meter height with flow rate of 2 L min$^{-1}$. An intermediate pressure lens (IPL) is used as aerodynamic lens allowing transmission of particles in the PM$_{2.5}$ fraction (Xu et al., 2017). The instrument uses a capture vaporizer (CV) to increase the particle collection efficiency (CE) compared to standard vaporizer (SV) (Jayne and Worsnop, 2016). By having a narrow entrance, the CV increases the particle collision events and thus increases the contact with the hot vaporizer surface, minimizing particles that bounce without evaporation (Hu et al., 2017) and resulting in higher CE. Consequently, however, the fragmentation patterns are shifted towards smaller ion masses due to additional thermal decomposition (Hu et al., 2017; Xu et al., 2017; Zheng et al., 2020). The average flow rate in the sample line of the instrument is 1.22 cm$^3$ s$^{-1}$ (0.07 L min$^{-1}$). Combining the PM$_{2.5}$ cut cyclone, PM$_{2.5}$ aerodynamic lens, and CV allows the ToF-ACSM to be a PM$_{2.5}$ measurement (Xu et al., 2017).

The ToF-ACSM provides unit mass resolution (UMR) mass spectra with 10-minute time resolution which are analyzed using Tofware v3.2 in Igor Pro 8. The fractions of measured UMR signals were assigned to individual aerosol species using the fragmentation table (Allan et al., 2004). On-site calibrations are performed to determine the ionization efficiencies of the chemical species. The calibrations of ionization efficiency (IE) and relative IE (RIE) were performed following the procedures described in previous studies by using 300–350 nm pure ammonium nitrate (NH$_4$NO$_3$) and ammonium sulfate ((NH$_4$)$_2$SO$_4$) dissolved in aqueous solution. The calibration gives IE value of 250.0 ions pg$^{-1}$ for nitrate (NO$_3$), and RIE values of 1.40, 1.67, 1.30, and 3.35 for organics (Org), sulfate (SO$_4$), chloride (Cl), and ammonium (NH$_4$), respectively. . The detection limits (measured similarly to Fröhlich et al., 2013) at 10-minute time resolution for this ToF-ACSM operating at Cabauw (a relatively polluted site in central Netherlands) are 0.38 μg m$^{-3}$ for Org, 0.12 μg m$^{-3}$ for NH$_4$, 0.07 μg m$^{-3}$ for NO$_3$, 0.11 μg m$^{-3}$ for SO$_4$, and 0.09 μg m$^{-3}$ for Cl.

In addition to the aerosol measurements by the ToF-ACSM, ambient sulfur dioxide ($SO_2$) concentrations were obtained from the open-source data of Landelijk Meetnet Luchtkwaliteit (LML, https://www.luchtmeetnet.nl), measured at the same location. Ammonia ($NH_3$) concentrations were obtained from measurements in Zegveld-Oude Meije station, 20 km to the north of Cabauw station (see Fig. 1a), also acquired from LML.

### 2.2.2. Particle size distribution measurements

The particle size distribution measurements were conducted using a TROPOS-SMPS. The instrument has been detailed in other work (Wiedensohler et al., 2012). Ambient air was sampled using a stainless-steel inlet equipped with $PM_{10}$ size-cut cyclone and Nafion dryer at 4.5-meter height sampling with a flow rate of 16.7 L min$^{-1}$. The SMPS inlet was placed approximately 3 m lateral distance from the ACSM instrument inlet. The instrument consists of a Vienna-type differential mobility analyzer (DMA) and a butanol-based TSI condensation particle counter (CPC) 3750. The flow rate in the instrument
is 1.0 L min$^{-1}$. The TSI CPC 3750 has the collection efficiency of 100% at the first selected and reported size of 10 nm.

  The raw dataset was processed using a linear multiple charge inversion algorithm to derive the particle number size distribution (PNSD or dN/dlog($D_p$)) (Wiedensohler et al., 2012; Pfeifer et al., 2014). The MPSS inversion algorithm version 2.13 was utilized to obtain final PNSD from the raw dataset. The final PNSD has 5-minute time resolution and covers 71 geometric
mean diameters ($D_p$) from 8 nm to 853 nm. The particle number concentrations (dN) for individual $D_p$ were then calculated by multiplying PNSD with dlog($D_p$) values.

### 2.3. Positive matrix factorization (PMF)

  The 10-minute average matrices of UMR organic fragment mass spectra with mass-to-charge ratio (m/z) 12 to 120 were combined with the inorganic species average mass concentrations (i.e., ammonium ($NH_4$), nitrate ($NO_3$), sulfate ($SO_4$), and
chloride (Cl)), and the 10-minute average particle number concentrations (dN) in 18 particle diameter size bins from 71 geometric mean diameters ($D_p$) to generate hybrid input data matrices for PMF analysis. Each organic fragment m/z, species concentration, and size-binned particle concentration is treated as an individual variable in the PMF.

  The values and errors of organic fragment mass spectra and inorganic mass concentration variables, and the minimum error
(minErr) of all species were generated by Tofware v3.2 in Igor Pro 8. The 10-minute resolution particle size dataset was obtained from the 5-minute resolution particle number concentration described in the Sect. 2.2.2. The particle number concentrations are categorized into 18 size bin variables (8–10 nm; 10–13 nm; 13–16 nm; 16–20 nm; 20-25 nm; 25–32 nm; 32–40 nm; 40–51 nm; 51–65 nm; 65–83 nm; 83–107 nm; 107–140 nm; 140–185 nm; 185–249 nm; 249–342 nm; 342–481 nm; 481–691 nm; and 691–853 nm). Each size bin contains the sum of four concentration points (except for the last bin
containing only three concentration points), then averaged to 10 minutes. We use larger bin sizes for the larger diameters

because larger particles occur less frequently. The errors for each size bin are taken to be the standard deviation of the raw data.

We performed the analysis using the PMF Evaluation Tool (PET) v3.08 (Ulbrich et al., 2009) in Igor Pro 8. The details of applying positive matrix factorization (PMF) to aerosol mass spectrometry datasets have been discussed elsewhere (Paatero and Tapper, 1994; Paatero, 1999; Ulbrich et al., 2009). Prior to analysis before the PMF input matrix preparation, the variable values and errors of species mass concentrations and particle number concentrations were downweighted in reference to the highest average peak of the original organic mass spectrum, m/z 44 ($f_{44}$). During PMF input matrix preparation in PETv3.08, the m/z 44, 28, 18, 17, and 16 signals in the organic mass spectrum are also downweighted as provided by the procedure to account for duplicated information of m/z 44 in the organic mass spectrum (Ulbrich et al., 2009). The details of PMF variable downweighting can be found in Sect. S3.

The first step of the factor analysis was identifying the optimum number of factors ($p$) by running unconstrained experiments using 2 to 8 factors and varying the seed value (min = 0; max = 20; delta = 1) to pick different initial values for the PMF algorithm. The optimum $p$ is selected by considering the lowest residuals and local minima ($Q/Q_{exp}$) of the PMF solutions. Alongside the local minima, we considered whether all the factors are environmentally reasonable and unique mainly based on their particle size distribution and chemical composition. After the optimum $p$ and seed value are chosen, the rotationality of the best PMF solution is explored by varying the rotation ($f_{peak}$) value (min = -1, max = +1, delta = 0.2). Bootstrapping runs with 100 iterations on the chosen PMF solution were performed to estimate the uncertainty in the factor profile variables and time series, ensuring the robustness of the solution.

To determine the organic and inorganic composition in each PMF factor, the particle size distributions are removed from the factor profile. The total organic mass fraction is considered as the sum of organic fragments fraction from m/z 12 to 120, while the inorganic mass fractions are upweighted back and taken as $NH_4$, $NO_3$, $SO_4$, and Cl mass fractions. The final fraction of each species is determined by dividing the species mass fraction with the total organic and readjusted inorganic mass fraction.

## 2.4. Wind analysis

To analyze the factors using wind variables, we investigate the prevailing wind for several pollution episodes observed in the dataset. Bivariate polar plots are generated for factor reconstructed mass concentration derived from PMF analyses and mass concentration for each ACSM species in each period using the "Openair" package in the "R" environment (Carslaw and Ropkins, 2012). The wind parameters are obtained from co-located measurement of 10-meter wind direction data acquired from KNMI.

## 3. Results and Discussion

### 3.1. Mean bulk atmospheric chemical composition across periods

We hypothesize that the mean bulk atmospheric chemical composition influences how the chemical species are distributed across the PMF factors. Therefore, we discuss this topic before the PMF solutions. To compare the mean bulk composition among periods, we choose the springtime period (May) as reference. The mean concentrations of atmospheric species and the species percentages in the bulk atmosphere are summarized in Table S1. In summer (June), we observe roughly a doubling in aerosol concentration for all IA and an increase with a factor 2.6 in OA compared to spring (May). In autumn (September), the particle concentrations decrease again, with a relatively larger decrease for sulfate.

**Table 2.** Mean bulk atmospheric chemical composition in the three periods, summarized as the values of total aerosol mass loading in µg m$^{-3}$, ion balance ratio ($NH_{4\_bal}$) from linear regression, and mean organic-to-inorganic ratio ($m_{Org}/m_{IA}$). A more detailed information about each chemical species can be seen in Table S1.

| Mean value | May 2021 (spring) | Jun 2021 (summer) | Sep 2021 (autumn) |
|---|---|---|---|
| Bulk aerosol composition[a] <br> ■ Org  ■ SO$_4$ <br> ■ NH$_4$  ■ Cl <br> ■ NO$_3$ | 37.7% / 1.6% / 17.5% / 27.9% / 15.3% | 45.1% / 1.0% (Cl) / 16.2% / 24.2% / 13.4% | 42.4% / 0.9% (Cl) / 10.9% / 31.2% / 14.6% |
| Aerosol mass loading[c] | 6.60 µg m$^{-3}$ | 14.12 µg m$^{-3}$ | 5.15 µg m$^{-3}$ |
| $NH_{4\_bal}$[b] | 0.997 ± 0.001 | 0.986 ± 0.001 | 1.066 ± 0.001 |
| $m_{Org}/m_{IA}$[a] | 0.61 | 0.82 | 0.74 |

[a] mass concentration, [b] molar concentration, [c] total mass of aerosol detected by ToF-ACSM ($m_{Org}+m_{NO_3}+m_{NH_4}+m_{SO_4}+m_{Cl}$)

The ion balance ratio, or also called ammonium balance ($NH_{4\_bal} = n_{NH_4}/(n_{NO_3}+2\times n_{SO_4}+n_{Cl})$), is the ratio between the measured ammonium ($n_{NH_4}$) and the total ammonium required to neutralize the major anions ($n_{NO_3}+2\times n_{SO_4}+n_{Cl}$). The ratio illustrates the excess of atmospheric ammonium (cation), or nitrate (anion), and other possibilities based on aerosol chemistry (see Sect. S2 for details). Ambient aerosol is normally charge-balanced, meaning that the major cation ($NH_4^+$) and major anion species ($NO_3^-$, $SO_4^{2-}$, and Cl$^-$) should have roughly one-to-one molar ratio ($NH_{4\_bal} \approx 1$). Among three periods analyzed, the ion balance ratio was found to be close to unity for all periods. This infers that the bulk aerosol charge is fully neutralized.

We introduce mean organic-to-inorganic mass ratio ($m_{Org}/m_{IA}$) to quantitatively compare bulk OA and IA composition across seasons. Based on this ratio, in summertime (June), we have composition richer in organics compared to spring and autumn. This difference is likely due to increasing biogenic emissions of volatile organic compounds (VOCs) in summertime, and higher temperature-induced increases in anthropogenic VOC concentrations.

## 3.2. Identification of PMF factors

From the unconstrained experiments using the combined ACSM-SMPS matrix, the best PMF solution was found to have 4 factors for May 2021 (Fig. 2), June 2021 (Fig. S1), and September 2021 (Fig. S2). The determination of PMF solution is detailed in Sect. S3. The resulting hybrid PMF solution matrix is split into organic mass spectrum, species mass concentrations, and particle number concentration bin matrices for the ease of presentation. The fragments of m/z below 20 are included in the PMF analysis but are not shown in the figures as they do not convey information for the spectrum interpretation. The signal factor axes for inorganic composition and particle size distribution are rescaled and fixed to allow comparison across factors. As each period encompasses around one month of time series data, the factors discerned by the hybrid PMF analyses show typical average aerosol composition during each period, rather than individual pollution episode profiles that may vary over time. Similarities and differences of factors across months will be discussed further below.

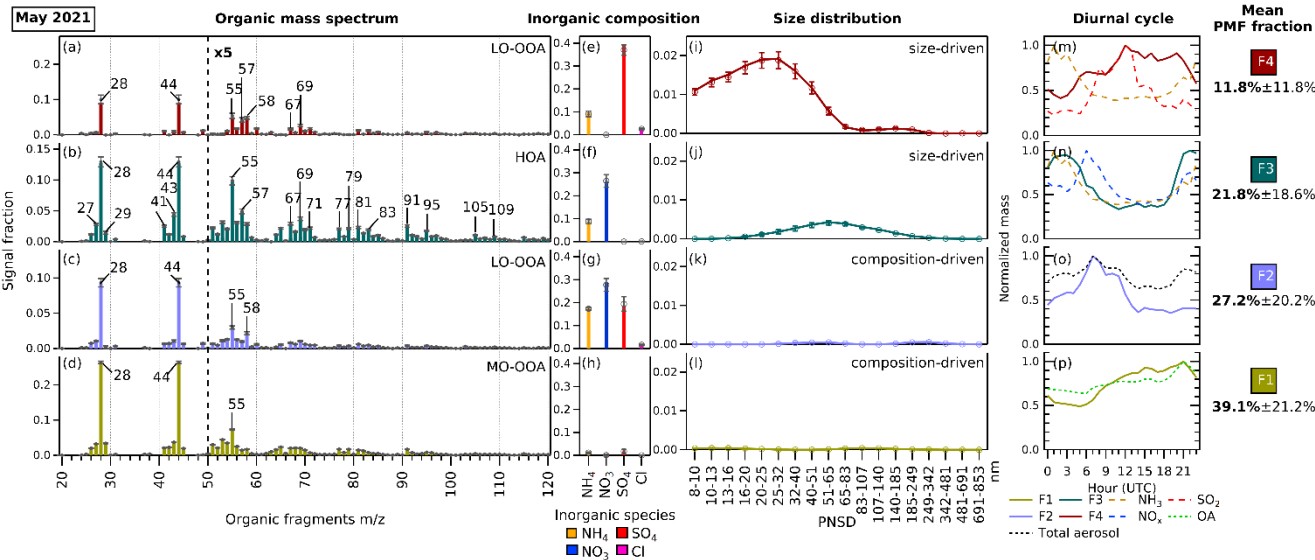

**Figure 2.** The profiles of 4-factor PMF solution from the combined ACSM-SMPS dataset in May 2021. Each factor is split into three matrices with their own rescaled signal fraction axes. The error bars in each variable represents standard deviation generated by performing bootstrapped run of the solution. The panel **a-d** shows the organic fragment mass spectrum from m/z 20 to 120 from ACSM (m/z < 20 not included). The panel **e-h** shows the ACSM standard inorganic aerosol species concentrations (ammonium ($NH_4$), nitrate ($NO_3$), sulfate ($SO_4$), and chloride (Cl)). The panel **i-l** shows the particle size distribution profiles from the SMPS. On the panel **m-p**, the diurnal cycles of the factors and related species are depicted. The mean PMF fractions and their standard deviations are shown indicating the mean contribution of each hybrid PMF factor to the "total variable reconstruction" by PMF throughout the period. Note that the standard deviations shown here indicate real variability in contribution of each factor, and not uncertainty. The factors in May 2021 are assigned as: (F1) MO-OOA, (F2) $NH_4+NO_3+SO_4+LO$-OOA, (F3) size-driven $NH_4+NO_3+HOA$, and (F4) size-driven $NH_4+SO_4+LO$-OOA. Similar figures for June and September 2021 can be found in Fig. S1 and Fig. S2.

### 3.2.1. Factor particle size distributions and composition

Two factors have particle size profiles associated with specific diameter subranges, which we interpret as related to NPF and growth. We therefore call these factors "size-driven". The size-driven factors resolved from the analysis possess similarities

in composition across months, where the factor associated with the smallest particle sizes is associated with bulk composition of ammonium sulfate aerosol (F4) while the larger sizes is linked to the bulk composition of ammonium nitrate aerosol (F3). The other two factors are unrelated to specific particle size and therefore called "composition-driven" factors, consisting of: (F2), an OA and IA mixed factor, and (F1), an OA-dominant factor. Both composition-driven factors can be seen as the representatives of bulk atmospheric aerosol composition. We can summarize that the NPF and growth follow the pathway starting from F4 and F3 into F2 and F1 (bulk aerosol composition), likely through processes such as condensation of gaseous precursors ($SO_x$, $NH_3$, $NO_x$, and VOCs and their reaction products) or particle coagulation (see Fig. 3). We note that this does not imply that all aerosol growth proceeds sequentially through these four factors; a more detailed discussion of possible NPF and growth pathways is found below in Sect. 3.3.4.

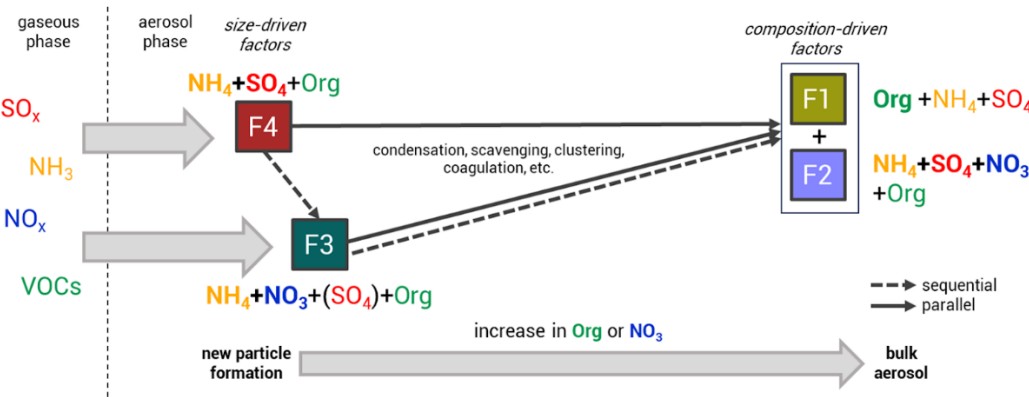

**Figure 3.** Potential relationships between the size-driven factors (F4 and F3, linked to NPF and growth) with the composition-driven factors (F2 and F1, linked to the bulk atmospheric aerosol composition), illustrating multiple possible aerosol growth pathways. From F4 (mainly ammonium sulfate) and F3 (mainly ammonium nitrate), particles can grow into F2 (OA and IA mixed) and/or F1 (OA-dominant), either sequentially (dashed), in parallel (solid), or combined. The particle formation and growth occur through condensation of gaseous precursors or particle coagulation. An increase in organics and $NO_3$ in the bulk composition is observed as particles progress along these pathways.

### 3.2.2. Factor organic profiles

The organic mass spectrum can be used to obtain information regarding the degree of oxidation, which can be related to atmospheric aging of each factor profile. In general, OA can be categorized into two types: primary organic aerosols (POA), and secondary organic aerosols (SOA). Oxygenated organic aerosols (OOA) are often considered to be SOA, while other OA profiles are generally considered POA (Chen et al., 2022).

OOA are characterized by relatively high m/z 28 ($f_{28}$) and m/z 44 ($f_{44}$) fragment signals, originating primarily from $CO^+$ and $CO_2^+$ fragments of carboxylate groups in organic compounds, produced by thermal decomposition inside the ACSM vaporizer (Alfarra et al., 2004). The $f_{44}$ fragment is often related to a high degree of oxidation and photochemical ageing (Alfarra et al., 2004; Ng et al., 2010). It is also important to note that the ACSM used in this paper has a CV instead of an SV inlet (see Sect.

2.2.1), which is known to produce higher $f_{44}$ values due to enhanced thermal decomposition (Hu et al., 2017; Zheng et al., 2020). The m/z 43 ($f_{43}$) fragment is characteristic for both oxygenated organic compounds ($CH_3CO^+$) and saturated hydrocarbon compounds ($C_3H_7^+$). Thus, factors with higher $f_{44}$ and lower $f_{43}$ values are understood to be more oxidized while lower $f_{44}$ and higher $f_{43}$ values implies that the factor is less oxidized. OOA may appear in more than one factor in a PMF solution and thus it is common to distinguish less oxidized-OOA (LO-OOA), typically associated with higher volatility organics, and more oxidized-OOA (MO-OOA), typically associated with lower volatility organics. To assess the variation in OOA oxidation level, the triangle plot (Ng et al., 2010) is normally used to compare $f_{44}/f_{43}$ values among resolved OOA factors in the PMF solution (see Fig. S6). OOA generally increase throughout the afternoon as its formation is photochemically driven (Hu et al., 2016; Sun et al., 2016) and accumulate in the evening due to the shallow nocturnal boundary layer. In the morning, the concentration decreases as clean airmasses introduced into the rising boundary layer diluting existing aerosol concentrations (Stull, 1988).

POA consists of various sources which can be identified from the appearance of certain fragmentation patterns in the organic mass spectrum, diurnal cycle, as well as correlation with other measurements. Some of the most common POA from PMF analysis are hydrocarbon-like organic aerosols (HOA), biomass burning organic aerosol (BBOA), cooking organic aerosols (COA), and coal combustion organic aerosol (CCOA) (Chen et al., 2022). POA have similar characteristics of alkyl and alkenyl fragments ($C_nH_{2n+1}^+$: m/z 29, 43, 57, 71, … and $C_nH_{2n-1}^+$: m/z 27, 41, 55, 69, …). HOA as a type of POA is often correlated with anthropogenic combustion pollutants, such as nitrogen oxides ($NO_x$) and black carbon from vehicular emission (Alfarra et al., 2004; Mohr et al., 2009; Zheng et al., 2020). Other POA profiles are distinguished by looking at certain fragments (e.g. m/z 60 and 73 for BBOA (Schneider et al., 2006; Weimer et al., 2008; He et al., 2010), m/z 55 for COA (He et al., 2010; Mohr et al., 2012), m/z of larger fragments related to polycyclic aromatic hydrocarbons (PAHs) for CCOA (Hu et al., 2013)). While CV increases the $f_{44}$ values and smaller organic fragments due to enhanced thermal decomposition, larger fragments become underestimated (Hu et al., 2017; Zheng et al., 2020) and therefore the differences between POA factors becomes more subtle.

The PMF analyses in this study resolved one POA factor (as HOA factor) and three SOA factors (two LO-OOA factors and one MO-OOA factor) across periods, where each organic profile has its corresponding IA composition and size distribution. Among size-driven factors, the factor related to the smallest particle sizes (F4) has OA assigned as LO-OOA, while the OA associated with the second size-driven factor (F3) is assigned as HOA. In the composition-driven factors, the OA+IA mixture factor (F2) resolves an LO-OOA profile while the OA factor (F1) resolves an MO-OOA profile.

### 3.3. Size-driven factors (F4 and F3)

Using the hybrid ACSM-SMPS datasets, two size-driven factors emerge from the PMF analyses as F4 and F3 (see Fig. 4). These factors are considered "size-driven" due to the approximately normally distributed particle concentrations in a specific sub-range of diameter. The two factors display different particle size clusters increasing in diameter from F4 to F3.

New particle formation (NPF) events are characterized by the rapidly increasing particle number concentration below 20 nm followed by particle growth, creating nearly vertical aligned peaks in particle number concentration plotted against time (Heintzenberg et al., 2007; Kerminen et al., 2018). By comparing the time series of particle size distribution (dN/dlogD$_p$), total mass loading, and PMF mass fraction (see Fig. 5 (May) and Fig. S3 (June and September)), we can observe that the episodes during which the size-driven factors' fraction increase occur when the total aerosol mass concentration is relatively low. This is to be expected, as during these periods, the condensational sink, which would compete by scavenging low-volatility gases or small particles, is reduced. If we zoom into the time series, the NPF growth shapes appear during episodes that are dominated by F4 and/or F3 (see Fig. S4).

The reconstructed PMF masses show the influence of sunlight and temperature on NPF events. The average PMF mass fraction of the size-driven F4 is larger in summertime (June) compared to other periods (see Fig. 4) due to higher mean radiation and temperatures (see Table 1). In summer (June), F4 accounts in average 14.9% of total reconstructed PMF mass while in spring (May) and autumn (September), it only represents 11.8% and 7.8%, respectively. The more frequent appearance of NPF growth events during summer can be seen in Fig. S3a-c. Other studies have likewise found the occurrence of NPF events is generally favored in high radiation (Modini et al., 2009; Peltola et al., 2022) and warmer temperatures (Jokinen et al., 2022; Peltola et al., 2022). This is because solar radiation provides the UV radiation that promotes photochemical reactions and turbulent motions needed to form new particles (Wehner et al., 2015; Dada et al., 2017; Kerminen et al., 2018; Sellegri et al., 2019).

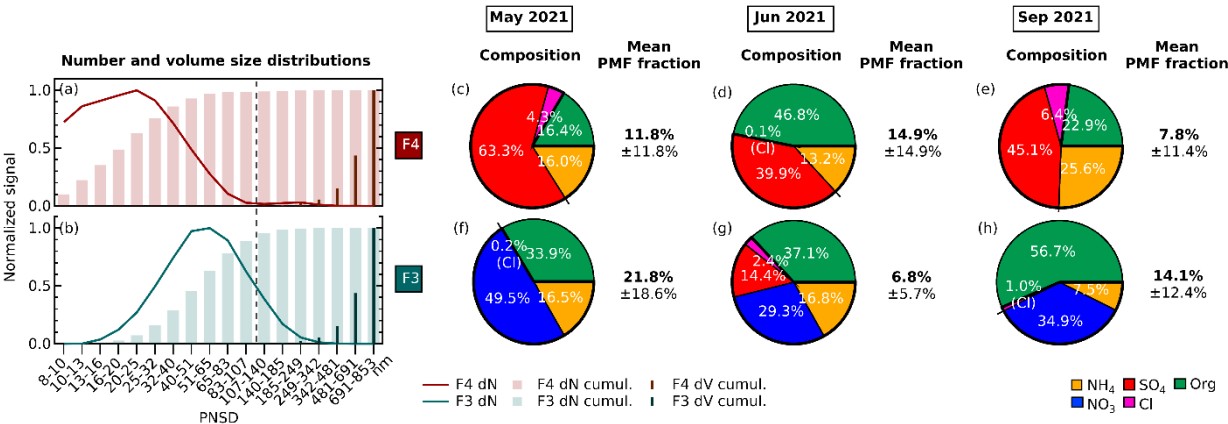

**Figure 4. (a-b)** Normalized average particle size distribution of two size-driven factors of F4 (maroon) and F3 (turquoise) across periods. The line plot shows particle number concentration (dN) fraction on each size bin. The thick histogram represents cumulative particle number concentration (dN) fraction as the particle size increases, while the thin histogram represents cumulative particle volume (dV = dN×(4/3)×π×(D$_p$/2)$^3$) fraction. The vertical dashed dark grey line divides the particle diameters where particles are transmitted with <50% efficiency on the left (diameter less than ~100 nm) and with at least 50% efficiency by PM$_{2.5}$ lens of ToF-ACSM on the right (diameter more than ~100 nm). **(c-h)** Pie charts showing mass percentage of each aerosol species contributing to each size-driven factor in May 2021, June 2021, and September 2021. Green represents organics (Org), orange represents ammonium (NH$_4$), dark blue represents nitrate (NO$_3$), dark red represents sulfate (SO$_4$), and pink represents chloride (Cl). F4 are dominated by ammonium sulfate while F3 are dominated by ammonium

nitrate. The mean PMF fractions and their standard deviations are shown, indicating the mean contribution of the factor to the total
reconstructed PMF mass and its variability over each month-long period.

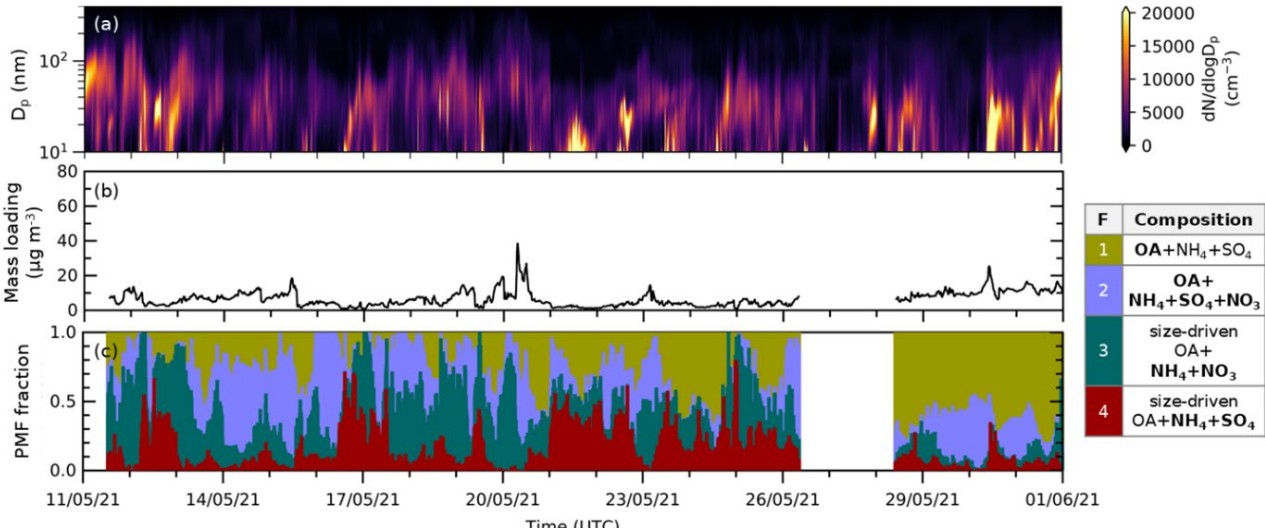

**Figure 5.** Time series of (a) particle size distribution ($dN/dlogD_p$) in cm$^{-3}$ with logarithmic scale in particle size obtained from SMPS measurements, (b) total mass loading calculated from ACSM species concentration (using Tofware) in µg m$^{-3}$, and (c) reconstructed PMF fraction (stacked) from analysis in May 2021. Episodes of F4 and F3 coincide with relatively low total aerosol mass conditions and high fine
particle concentrations. Similar figures for June and September 2021 can be found in Fig. S3.

### 3.3.1. Particle size distributions

F4 corresponds to the bulk composition when the particles in the nucleation mode are growing into Aitken mode size range, with modal size of 20-25 nm (see Fig. 4a). F3 is characterized by clustered particle sizes in the Aitken and accumulation mode region, with modal size of 51-65 nm (see Fig. 4b). The F4 and F3 mass loading shows a good correlation with particle number
concentration in the size bin of 20-25 nm and 51-65 nm, respectively, across periods (see Fig. S5). The size range differs slightly across months, but F4 always appears as the smallest particle size range among factors, which we therefore designate as the nucleation-mode factor, while F3 with larger particle size range is designated as the growth-mode factor.

Some concerns may arise because the ACSM and SMPS measure different particle size ranges, especially the smaller sizes
which are the major interest of this study. According to a study conducted using a ToF-ACSM in the same configuration as this study, the PM$_{2.5}$ lens in the ToF-ACSM transmits particles with vacuum aerodynamic diameter ($D_{va}$) between 100 nm and 3 µm with efficiency above 50%, decreasing to around 20% for $D_{va} \sim$ 55–60 nm (Xu et al., 2017). Meanwhile, the SMPS instrument samples particles with diameters ranging from 10 to 800 nm.  As we are mainly interested in elucidating the composition of NPF and growth in sizes finer than 100 nm, we address here the influence of this mismatch on the interpretation
of our PMF results.

In Fig. 4a, we observe that F4 mainly related to aerosol with sizes that have transmission efficiency <50% in PM$_{2.5}$ ToF-ACSM, while in Fig. 4b, only the size bins of 107–140 nm in F3 have transmission efficiency ≥50%. The small-sized particles will make a negligible contribution to the PM$_{2.5}$ mass and the larger particles will always dominate the particle volume size distribution, regardless of whether the finer particles are efficiently sampled or not by ACSM. Nevertheless, although the ACSM does not directly measure the finest particle composition, the factor still illustrates the bulk chemical composition that occurs during and favors the formation and growth of new particles. Moreover, the PMF tool does not recognize that the diameter bins are sequential, so the fact that particle diameters cluster in certain ranges in factors F4 and F3 is consistent with their identification as corresponding to NPF and growth.

### 3.3.2. Chemical composition

In all periods, the nucleation-mode F4 has ammonium sulfate as the major component (see Fig. 4c-e). The factor further consists of ammonium (13% to 26%), sulfate (40% to 63%), organic compounds (16% to 47%), and traces of chloride (0.1% to 6.4%). Organics are known to participate in particle formation and growth (Riipinen et al., 2012; Hodshire et al., 2016), while in this study, the mass percentage share between ammonium, sulfate, and organics of F4 depends on the mean bulk organic composition in each period (see Fig. 4c-e). The low mean bulk organic composition in springtime and autumn (May and September) leads to F4 being largely ammonium (16% and 26%) and sulfate (63% and 45%), followed by OA (16% and 23%) and chloride (4.3% and 6.4%). The organic-rich regime in summer (June, see Table 2) results in the increase of OA in F4 (47%) and less ammonium, sulfate, and chloride (13%, 40% and 0.1%, respectively). F4 represents from 9.5% up to 14.3% of total reconstructed PMF mass in the solution with the highest being during summer when there is more contribution from organic masses. On the other hand, the growth-mode F3 is composed of mainly ammonium nitrate aerosol (see Fig. 4f-h). The factor further composed of ammonium (8% to 17%), nitrate (29% to 50%), organic compounds (34% to 57%), and traces of chloride (0.2% to 2.4%). In contrast to other months, the PMF analysis also resolves variations in F3 containing sulfate (14%) during summertime (June).

Overall, we interpret these results to indicate that sulfate is a key component of nucleating particles during NPF events. When the mean bulk organic concentration is high and more oxidized (e.g., summertime), it participates more abundantly in particle nucleation. While sulfate is key to nucleation, nitrate plays a more important role in particle growth (see Sect. S2).

### 3.3.3. Organic profiles

The organic mass spectrum profile from each size-driven factors and their diurnal cycles in each period are shown in Fig. 6. Across seasons, LO-OOA is part of bulk composition related to nucleation-mode particles. The factors are assigned as LO-OOA due to their $f_{44}/f_{43}$ values compared to other OOA factors (see the triangle plot in Fig. S6). The LO-OOA F4 profile resolved in this study is comparable to LO-OOA resolved in other aerosol mass spectrometry studies using CV (Zheng et al., 2020; Joo et al., 2021), although fragments with m/z > 50 are less prevalent. Several aerosol chamber experiments have

reported that lower volatility and highly oxygenated organic molecules from biogenic and anthropogenic organic precursors play a dominant role in new particle formation and growth (Schobesberger et al., 2013; Ehn et al., 2014; Riccobono et al., 2014; Tröstl et al., 2016; Mohr et al., 2019; Pospisilova et al., 2020; Zhao et al., 2021). In this study, however, we surprisingly observe LO-OOA rather than MO-OOA associated with nucleation. This could imply that organic compounds with less oxygenation are more abundant and condense on freshly nucleated particles in this region, or that the ToF-ACSM delineation between LO-OOA and MO-OOA does not directly correspond to volatility in this case.

In terms of diurnal cycle, the F4 mass loading does not follow the typical LO-OOA pattern, but rather together with the formation of ammonium sulfate is responsible for the increase in new particle loading during the day, as seen in the diurnal cycles (Fig. 6g,i,k). The organic-rich regime in summer (see Table 2) combined with higher mean temperature also favors the abundant production of semi-volatile OA which can condense onto newly formed particles, leading to the increase of OA to 47%, compared to 16% in spring and 23% in autumn (see Fig. 4).

F3, seen as the growth-mode factor related mainly to organic and ammonium nitrate bulk composition, has an HOA-like organic profile. The factor is assigned as HOA due to the alkyl and alkenyl fragments that are abundant (m/z 27, 29, 41, 43, 55, 57, 69, and 71). The diurnal cycle of F3 in this study shows similarity with general non-urban HOA diurnal patterns, with increased loading at nighttime (see Fig. 6h,j,l), contrary to typical diurnal pattern of HOA in urban sites with peaks during morning and evening rush hour (Chen et al., 2022). It suggests that the growth-mode F3 may be related to transported vehicular emissions from urban areas and/or local primary organic emissions. Organic nitrates can also be formed from reaction between $NO_x$ with less oxygenated VOCs through $NO_3$ or alkyl peroxy radical chemistry (Berkemeier et al., 2016). The diurnal pattern of F3 is consistent with this organic nitrate formation, followed by condensation onto the newly formed particles as the temperature lowers at night. During the summer, a small increase in F3 mass loading during the daytime also can be observed, hinting at enhancement of daytime organic nitrate formation in the hottest and sunniest period.

During the summer, the values of $f_{28}$ and $f_{44}/f_{43}$ as oxygenated organic markers in F4 are higher compared to other seasons, while they are almost absent in F3. In the chemical composition discussed in Sect. 3.3.2, sulfate also makes appearance in F3 in summer, unlike other seasons. We hypothesize that the PMF solution did not resolve the two size-driven factors in summer (Jun) similarly to other seasons, and therefore the aerosol composition is more mixed between NPF and growth particles. Mathematically, this is reflected in the local minima of the chosen PMF solution. Despite being the lowest minimum in the PMF solution space, $Q/Q_{exp}$ in June scores lower compared to May and September (see Table S3).

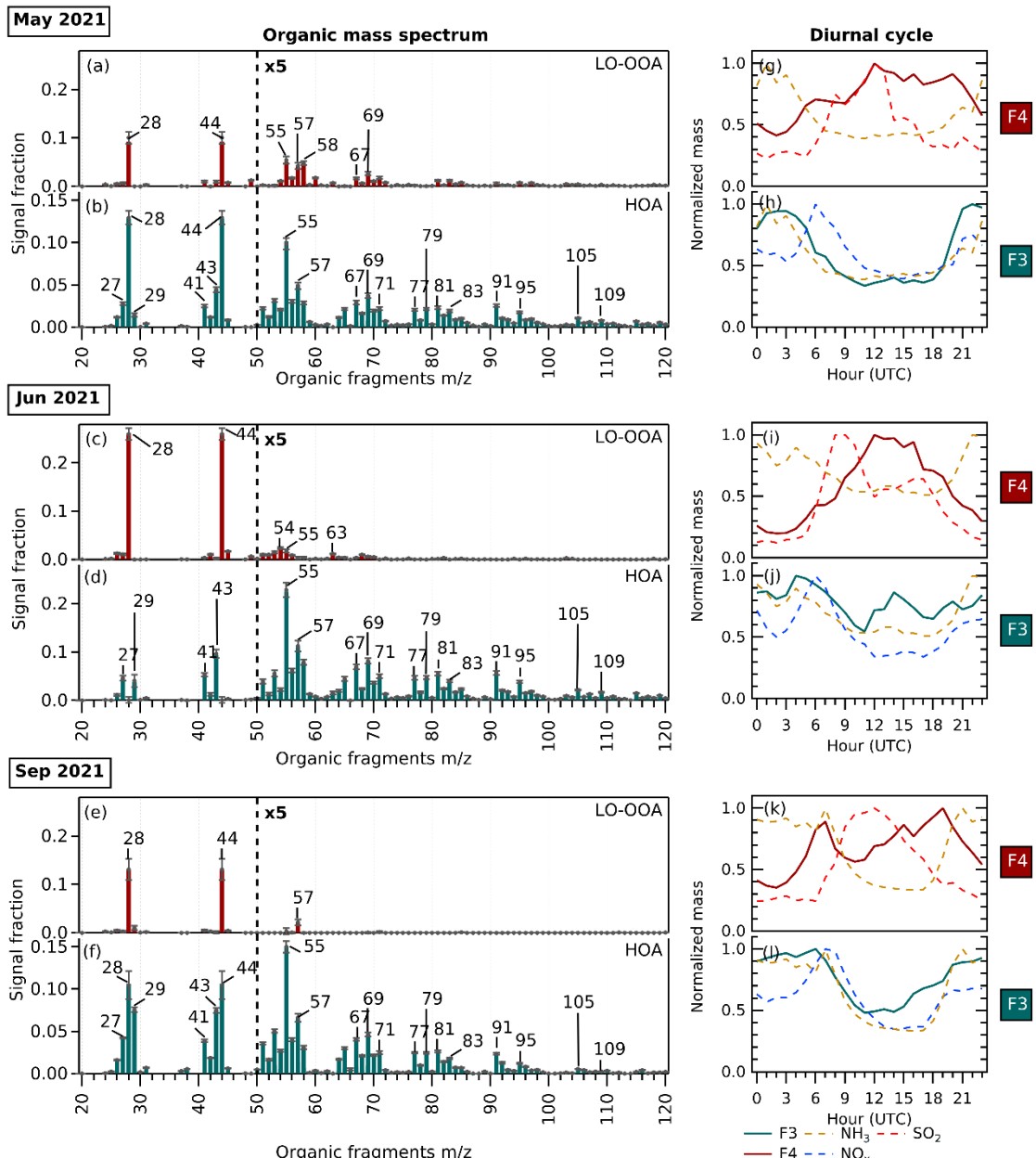

**Figure 6. (a-f)** Organic mass spectrum from m/z 20 to 120 (m/z < 20 not included) of F4 (maroon) showing LO-OOA factor profiles and F3 (turquoise) showing HOA factor profiles in May 2021, June 2021, and September 2021. The error bars in each m/z were generated from bootstrap run. **(g-l)** Diurnal cycles of corresponding factors and potential precursor gases. The diurnal cycles of F4, mainly composed of ammonium sulfate, are shown together with its precursors, $NH_3$ and $SO_2$. The diurnal cycles of F3, mainly composed of ammonium nitrate, are shown together with its precursor, $NH_3$ and $NO_x$.

### 3.3.4. New particle formation (NPF) and growth pathway

The NPF events shown by the time series of particle number size distribution and size-driven factor loading reveal that particle formation and growth takes around 6 to 12 hours (see Fig. S4). The high occurrence of ammonium sulfate and oxidized organic molecules in the aerosol phase observed during nucleation-mode F4 episodes marks the beginning of NPF. This is related to ammonium sulfate formation from the reaction between ammonia and sulfuric acid, and uptake of oxygenated organic compounds. We can consider F3 as a "sequential" pathway of F4 growing in size (see Fig. 3); this sequential nature is observed in some NPF events shown in Fig. S4, when F3 peaks after F4. F4 grows into F3 when nitric acid and/or organic nitrates and hydrocarbon-like semi-volatile organic compounds are dominant in the aerosol composition.

One might argue that F3 cannot be considered the successor of F4 as it does not contain any sulfate (for May and September), which should remain from initial nucleation. Another possible explanation is that F3 emerges directly from ammonium nitrate as a "parallel" nucleation pathway (see Fig. 3). Other studies have observed this nucleation mode to occur very rarely and only in the free troposphere, at lower temperature and very clean air conditions, through reaction between nitric acid and $NH_3$ (Höpfner et al., 2019; Wang et al., 2020). In a "combined" hypothesis, F3 emerges from F4, but the particles rapidly favor the pathway of growing mainly by ammonium nitrate condensation to the particle phase. The simultaneous pathway of F4 and F3 growth can be observed in some NPF events in Fig. S4. This process leads to the negligible amount of sulfate and abundance of ammonium nitrate during particle growth, hence sulfate mass being unresolved in the F3 composition (for May and September). Chamber experiments and theoretical studies support this interpretation of NPF occurring with only minor involvement of sulfate (Liu et al., 2018; Wang et al., 2020, 2022). The chemistry of ammonium sulfate and nitrate aerosol formation is discussed in more detail in Sect. S2.

### 3.3.5. Relationship of new particle formation with wind variables

To study the relationship between wind variables and new particle formation in the rural site of Cabauw, wind analyses were done using bivariate polar plots of size-driven F4 and F3 by wind speed and wind direction (see Fig. 7). We observe that nucleation-mode F4 are correlated mainly with air masses transported from southwesterly-westerly sector, and sometimes from the easterly sector. These wind sectors supply sulfate, ammonium and organics (Fig. S8), and their precursor gases that determine the main composition of F4 (see Fig. S9). Westerlies represent a source of sulfate, which mainly comes from sulfur oxides ($SO_x$) emission along the waterway of Rotterdam's harbor, the busiest port in Europe. The sulfate in air transported to Cabauw from the northern to eastern sector may arise from $SO_x$ precursor from other urban, shipping, industry centers (e.g., Amsterdam city and port, Utrecht city), and power plants to the site (see Fig. 1) (Henschel et al., 2013; Fioletov et al., 2016; Ledoux et al., 2018). The supply of ammonium for new particle condensation through $NH_3$ emission comes from the agricultural practices that take place around the Cabauw site, with tendency of higher $NH_3$ and ammonium from the southern

sector. The easterlies extending to north are also sources of VOCs coming from the forested nature areas in the provinces Utrecht and Gelderland, which are subsequently transformed into SOA.

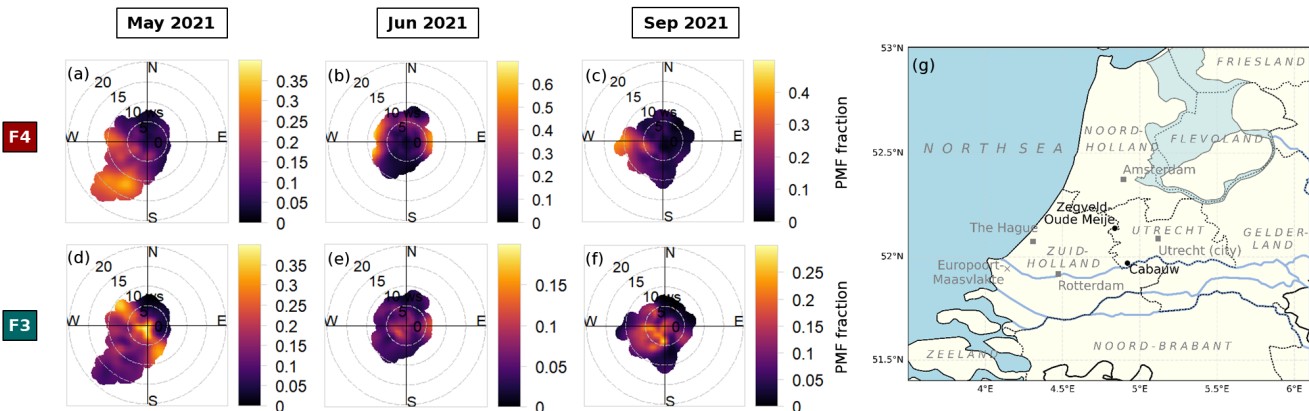

**Figure 7. (a-f)** Bivariate polar plots of size-driven factor mass fraction (F4 and F3) by wind speed and wind direction measured in Cabauw in May, June, and September 2021. We can observe that nucleation-mode particles (F4, **a-c**) fraction is largely correlated with air masses from the southwesterly-westerly sector where the urban and harbor areas of Rotterdam are located. The growth-mode particles (F3, **d-f**) fraction is distributed around the site and from southwesterly sector. Similar figures for composition-driven factors can be found in Fig. S7.

The different prevailing wind also affects the F4 composition and frequency. In spring (May) and autumn (September), new particle formation mainly correlates with winds from southerly and westerly directions (see Fig. 7a,c), and thus has less organic composition. In summer (June), winds coming from the east contribute to NPF events (see Fig. 7b), supplying more organics to the site. The abundant organics due to higher radiation and temperatures allow semi-volatile SOA to directly condense onto
newly formed particles, increasing NPF events and F4 mass fraction during summer (see Fig. 4d).

### 3.4. Composition-driven factors (F2 and F1)

The two composition-driven factors yielded by the PMF analyses are F2 and F1 (see Fig. 2 (May), Fig. S1 (June), and Fig. S2 (September)). We call these factors composition-driven because they are found across the size distribution, rather than in a specific size range. They collectively account for a large fraction of total reconstructed PMF mass (66% to 78%) and can be
considered as the result of size-driven factors further growth into the bulk aerosol. F2 and F1 are both representative of mean atmospheric bulk aerosol composition that are split into two different factors by PMF.

F2 is characterized by the presence of a mixture of OA and IA (see Fig. S10m-r). Based on the organic spectrum, the aerosol mixture can be characterized as LO-OOA, comparable to the LO-OOA profile found in other aerosol mass spectrometry studies
using CV (Hu et al., 2018; Zheng et al., 2020; Joo et al., 2021). It has similar $f_{44}/f_{43}$ to LO-OOA profile in F4 in average (see triangle plot in Fig. S6). In terms of diurnal cycle, the F2 mass loading does not follow the typical LO-OOA pattern. Normally,

LO-OOA would have a higher nighttime concentration and slight decrease during the day (Chen et al., 2022), but we found the diurnal cycle of F2 to be similar to the diurnal pattern of total aerosol mass loading in this study (see Fig. S10g,i,k). This finding suggests that F2 is the result of condensation of available semi-volatile chemical constituents over the course of the day as the continuation of NPF and growth, governed by the availability of both bulk IA and OA composition.

While F2 contains both OA and IA, F1 is dominated by OA (see Fig. S10m-r). It represents the continuation of NPF and growth enriched by organic compounds. F1 is composed mainly of organics (82% to 94%) and a trace amount of ammonium (2% to 6%) and sulfate (2% to 12%). A trace of nitrate aerosol (2%) can also be found in F1 in the summer (June). The abundance of m/z 44 fragments and high $f_{44}/f_{43}$ values (see Fig. S6) indicates that factor F1 represents aged SOA, resembling OOA profiles observed at the same site in previous studies (Mensah et al., 2012; Paglione et al., 2014; Schlag et al., 2016) and MO-OOA profiles observed in aerosol mass spectrometry with CV (Hu et al., 2018; Zheng et al., 2020; Joo et al., 2021). F1 exhibits a diurnal pattern consistent with MO-OOA, with concentration rising slightly across the day. It has a relatively stable concentration throughout the day because the factor is driven largely by long-range transport of aerosol (Kodros et al., 2020; Chen et al., 2022), but with slightly increasing concentration from morning to afternoon due to photochemical oxidation. Wind analysis suggests that the bulk SOA in F1 is related to airmasses arriving from the easterly sector that spans from north (~0°) to south (~180°) (see Fig. S7). To the east, the province of Gelderland is mostly covered by agricultural grass land and forested nature areas that emit VOCs, therefore increasing the amount of OA produced. Easterly wind directions may also contain the accumulated pollutants or VOCs from continental Europe, and therefore may contain a variety of OA from either biogenic or anthropogenic sources. Considering the organic profile, the high mass loading percentage, and the source regions across periods, we attribute F1 to background regional and continental OA.

## 4. Conclusions

In this work, we have shown that hybrid ACSM-SMPS PMF analysis can be used to determine the bulk chemical composition associated with new particle formation and growth. The analyses of three selected periods enable us to use the seasonality of the factor profiles, representing conditions of spring (sunny and warm) for May, summer (very sunny and hot) for June, and autumn (less sunny and warm) for September, as well as different prevailing winds, to attribute factor sources.

New particle formation episodes appeared when the total aerosol concentration was low, with key contributions from ammonium sulfate and oxygenated organic compounds across seasons, despite the high nitrogen emission in the Netherlands. This new particle formation and growth exhibits a diurnal pattern dominated by daytime formation that shifts to nighttime growth as the particle size increases. While sulfate promotes new particle formation, nitrate and semi-volatile organics are more influential in growth. The substantial contribution of nitrate and less oxidized organic aerosols to F3, and its shift to a nighttime peak in concentration, indicate that ammonium and organic nitrate condensing during the particle growth when the

temperature is lower. The organic-rich regime, higher mean radiation, and higher mean temperature in summer results in larger
contribution of oxygenated organic vapors in new particle formation.

New particle formation is most pronounced with winds from the southwest-west, and sometimes northeast. These directions supply precursors gases, with the westerlies bringing $SO_x$ from the port of Rotterdam, southwesterlies bringing $NH_3$ from agricultural emissions, and easterlies bring organic vapors from the forest and nature areas, The influence of the wind direction
can be clearly seen during the summer, where instead of southern and western winds, the prevailing winds were from the north and east and brought abundant organics, resulting in the rapid growth of large amounts of OA.

In sum, this combination of composition and size information into the statistical method of PMF, augmented by meteorological and gas-phase auxiliary data, provides a powerful tool to assess the factors that control aerosol production in a complex region,
heavily influenced by agricultural and industrial activities, alongside biogenic emissions of VOCs.

**Code availability**

The analysis and graphics are mainly generated using Igor Pro 8. The code utilized in the PMF analysis is part of the PMF Evaluation Tool (PET) v3.08 written as Igor procedures, available at https://cires1.colorado.edu/jimenez-group/wiki/index.php/PMF-AMS_Analysis_Guide#PMF_Evaluation_Tool_Software (Ulbrich et al., 2009). The map,
pollution roses, and particle size distribution time series are generated using Python 3.9 packages. The map is generated using "SciTools/cartopy v0.21.1" Python package, available at https://doi.org/10.5281/zenodo.7430317 (Elson et al., 2022). The pollution roses are generated using "windrose v1.8.1" Python package, available at https://doi.org/10.5281/zenodo.7465610 (Celles et al., 2022). The particle size distribution time series are plotted using open-source Python code written by Lee Tiszenkel, available at https://github.com/ltisz/Banana-Plot. The bivariate polar plots of concentrations are generated using
"Openair" package in the "R" environment, available at https://davidcarslaw.github.io/openair/ (Carslaw and Ropkins, 2012).

**Data availability**

The ACSM-SMPS datasets were collected as part of the Ruisdael Observatory Land-Atmosphere Interactions Intensive Trace-gas and Aerosol (RITA) campaign in May to September 2021 (https://ruisdael-observatory.nl) and available upon request. The gaseous phase species concentration can be retrieved from open-source data provided by Landelijk Meetnet Luchtkwaliteit
(LML, https://www.luchtmeetnet.nl). The meteorological variables during the period are provided by the Royal Netherlands Meteorological Institute (KNMI, https://www.knmi.nl) and available upon request.

**Author contributions**

JLF and FRN designed the research; RM, RH, BH, and XL collected the data; FRN analyzed the data; FRN and JLF wrote the manuscript draft; MCK, RH, and UD reviewed and edited the manuscript.

**Competing interests**

The authors declare that they have no conflict of interest.

**Acknowledgments**

This work has been accomplished by using data generated in the Ruisdael Observatory, a scientific infrastructure co-financed by the Dutch Research Council (NWO, grant number 184.034.015).The authors acknowledge valuable discussions with Doug Day, Donna Sueper, and Ian Chen. We would also like to express our gratitude to two anonymous reviewers for helpful comments.

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
