# Peer review of "What chemical species are responsible for new particle formation and growth in the Netherlands? A hybrid positive matrix factorization (PMF) analysis using aerosol composition (ACSM) and size (SMPS)"

_EGUsphere, 2023_

## Author Comment (AC1)

**Response to reviewers for "What chemical species are responsible for new particle formation and growth in the Netherlands? A hybrid positive matrix factorization (PMF) analysis using aerosol composition (ACSM) and size (SMPS)" by Nursanto, Farhan R; Meinen, R.; Holzinger, R.; Krol, Maarten C.; Liu, Xinya; Dusek, Uli; Fry, Juliane L. (Manuscript ID: EGUSPHERE-2023-554)**

We thank the reviewers for their thorough and constructive comments on our paper; we believe this revised manuscript is substantially improved thanks to their suggestions. To guide the review process we have copied the reviewer comments in black text, renumbered per reviewer to facilitate cross referencing. Our responses are in regular blue text. We have responded to all the reviewer comments and made alterations to our paper (**in bold text**).

We have made major revisions based on the reviewer's comments, including re-running and re-analyzing the PMF analysis, which gave a different number of factors in the optimum solution. (We note that the basic story of the paper is the same, with PMF identifying size-driven and bulk composition factors that give insight into species responsible for NPF and growth). However, these major revisions result in substantial changes to section order and figures. Therefore, for simplicity and to avoid confusion while reading the revised document, we do not provide a fully tracked change document. Here, we first summarize the changes made as a guide for reviewers and editors. Responses to individual reviewer comments are then listed below.

**Revisions in the main article**

- The main story in the abstract remains similar with some changes in the details
- Very minor changes in **"1. Introduction"**
- "**2. Methods and instrumentation**"
    - **2.1. Cabauw site and meteorological conditions:** Fig. 1 and Fig. 2 combined into Fig. 1, minor additional sentence describing the weather data.
    - **2.2. Measurement setup:** major revision, split into two subsection,
        - **2.2.1. Chemical species measurements:** ACSM instrument details and auxiliary gas measurements
        - **2.2.2. Particle size distribution measurements:** SMPS instrument details
    - **2.3. Positive matrix factorization (PMF):** major revision, more details on PMF input matrix preparation and analysis setup, resulting in changes in PMF solution
    - **2.4. Wind analysis:** very minor changes
- "**3. Results and discussion**"
    - 3.1. and 3.2. switched order
    - **3.1. Mean bulk atmospheric chemical composition across periods (previously 3.2)**
        - Removal of potassium (K) from the composition
        - Update on ion balance ratio interpretation
        - Removal of n SO4/n NO3 ratio
        - Revision on m Org/m NH4 to be m OA/m IA (organic vs inorganic)

- ○ **3.2. Identification of PMF factors**
  - ■ First paragraph from **"3.2.1. Factor particle size distributions and composition" (previously 3.1.1.)** is moved to serve as introduction for Section 3.2 with some modification
  - ■ Major revision in response to reviewer's comments on **"3.2.1. Factor particle size distributions and composition"** and **"3.2.2. Factor organic profiles"**, previously 3.1.1 and 3.1.2: PMF re-run with updated PMF input matrix and setting details, resulting 4-factor solution instead of 6-factor solution.
  - ■ Detailed changes can be found in the upcoming comment responses.
- ○ **"3.3. Size-driven factors (F4 and F3)":** major revisions, detailed in the upcoming comment responses)
  - ■ In the new 4-factor solution, size-driven factors are F4 and F3 (previously F6 and F5)
  - ■ Detailed changes can be found in the upcoming comment responses.
  - ■ Change in subsection arrangement:

| *Before revision* | *After revision* | *Comments* |
|---|---|---|
| 3.3.1 F6: nucleation-mode factor | **3.3.1 Particle size distribution** | Change in subsection arrangement corresponding more into PMF variables rather than discussing each size-driven factor separately |
| 3.3.2. F5 to F3: growth-mode factors | **3.3.2 Chemical composition** | |
| 3.3.3. Relationships with mean radiation and temperatures | **3.3.3 Organic profile** | |
| | **3.3.4 NPF and growth pathway** | New subsection |
| 3.3.4. Relationship of new particle formation with wind variables | **3.3.5. Relationship of new particle formation with wind variables** | Major revision |

- ○ **"3.4. Composition-driven factors (F2 and F1)":** major revision, **Section "S3. F1: Background OA factor" in the original document** is moved here.
  - ■ Detailed changes can be found in the upcoming comment responses.
- ● **"4. Conclusion":** the main story in the conclusion remains roughly similar with some changes in the details

- The figures and tables are updated as follows:

| Before revision | After revision |
|---|---|
| Figure 1. Map of a part of the Netherlands… | Combined into **Figure 1** |
| Figure 2. Wind rose plots for May, June, and September 2021 | |
| Figure 3. The profiles of 6-factor solution from the combined ACSM-SMPS dataset in May 2021. | Numbered as **Figure 2,** updated with 4-factor solution from new PMF run, |
| Figure 4. Pie charts showing mass percentage of each aerosol species contributing to each size-driven factor profile… | Numbered as **Figure 3,** updated with 4-factor solution from new PMF run, with additional plot of size distribution for each factor (F4 and F3). |
| Figure 5. Time series of (a) particle size distribution (dN/dlogDp) in cm-3 with logarithmic scale in particle size obtained from SMPS measurements, (b) total mass loading calculated from ACSM species concentration… | Numbered as **Figure 4,** updated with 4-factor solution from new PMF run,. |
| Figure 6. Normalized diurnal cycles in May 2021 of (a) the size-driven factors of F6 and F5… | Numbered as **Figure 5**, updated with 4-factor solution from new PMF run, with additional plot of factor's organic mass spectrum,. |
| Figure 7. Timeseries of (a) wind direction (WD) color-coded with wind speed (WS), and (b) reconstructed PMF fractions F6 and F5 (stacked) corresponding to nucleation-mode and first growth-mode particles in May 2021. | Numbered as **Figure 6,** information presented as bivariate polar plots instead of time series. Dataset from June and September 2021 that were presented as Figure S6 in the original document are also incorporated into this figure. |

**Revisions in the supplementary information (SI)**

- The line number reset from 0 instead continuing from the main article
- **"S1. Figures and tables":** table of figures and table of tables are removed.
- The figures and tables are updated as follows:

| Before revision | After revision |
|---|---|
| Figure S1-S4 | Updated according to the new PMF run results. |
| Figure S5. Normalized diurnal cycles in | **Moved to the main article as Figure 5**, |

| | |
|---|---|
| June and September 2021 of size-driven factors | incorporated with factor's organic mass spectrum

   ○ **Replaced by new information, Figure S5: bootstrapped time series of F4 and F3 plotted with the number concentration of particles in the size bin of 20-25 nm and 51-65 nm respectively** |
| Figure S6. Timeseries of wind direction (WD) color-coded with wind speed (WS), and reconstructed PMF fractions of size-driven factors | **Moved to the main article as part of Figure 6**, information presented as bivariate polar plots instead of time series

   ○ **Replaced by new information, Figure S6: the triangle plot for OOA assessment** |
| - | Added **Figure S7: Wind roses and (d-i) bivariate polar plots of composition-driven factor mass fraction (F2 and F1) by wind speed and wind direction** |
| Figure S7. Wind roses and bivariate polar plots of nucleation-mode particle precursor concentrations | Changed number into **Figure S9,** updated information |
| Figure S8. Wind roses and bivariate polar plots of ACSM-generated chemical species by wind speed and wind direction measured in Cabauw in May, June, and September 2021 | Updated to harmonize with the plot design of the new PMF results. |
| Figure S9. Diurnal cycles of F2 and total aerosol mass loading measured by the ACSM with associated composition across periods | Changed number into **Figure S10,** updated information and incorporated with factor's organic mass spectrum |
| - | **Figure S11-S13 added containing diagnostic plots of PMF analyses** |
| Table S1 | Updated according to the new PMF run results. |
| | **Table S2-S3 added containing diagnostic plots of PMF analyses** |

- **"S2. Atmospheric composition and aerosol formation"** is updated from the section **"S2. Determining composition regimes from bulk atmospheric composition"**
- The section is further divided into "**ion balance ratio/ammonium balance**" and **"ammonium sulfate and nitrate aerosol formation"** subsection, responding to reviewer's comments
- Removal of **"S3. F1: Background OA factor"**. The content is revised and incorporated into **"3.4. Composition-driven factors (F2 and F1)"** in the main article.
- Removal of "**S4. Correlation of chloride-sulfate and potassium-organic species**" due to removal of potassium from PMF analysis.
- Creation of "**S3. PMF analysis**" detailing **"PMF variables downweighting"** and **"Determination of PMF solution"**, which includes the addition of **Table S2, Table S3, and Figure S11-S13.**

**Reviewer #1**

Nursanto et al. utilize three months of observations collected in Cabauw, Netherlands, to investigate what aerosols control new particle formation (NPF). Using an Aerosol Chemical Speciation Monitor (ACSM) and scanning mobility particle sizer (SMPS) with positive matrix factorization (PMF), the authors found four distinct factors associated with the start and growth of new particles and two factors associated with background, large particles. The factors found by the authors were generally similar across the three different months. They generally found sulfate was associated with the beginning of an NPF event and nitrate was observed during the condensational growth. Further, organics were both associated with NPF and condensation. The authors also associate the events with different relative chemical composition of aerosol (organic rich, nitrate rich, sulfate rich, ammonium rich) and back trajectories for where the air masses originated.

Though this article is potentially of interest to the ACP community, and the authors have done a good job in setting up the scientific question and premise, there are many technical aspects and analysis that either needs further discussion and/or evaluation, which are detailed below, prior to publication to ACP.

Major

1) In general, much more details are needed in the measurements. There are many key details that are necessary in evaluating the science that have not been addressed. These include:

R1-1.1) What size particle lens was used? It is not clear if a PM1 or PM2.5 particle lens was used, which is both important in the flow rates for the cyclone and the aerosol diameter cut-offs (further discussed below).

A 2.5 µm size cut was used in both the cyclone and internal aerodynamic lens. We have added text to line 123-125 of the paper mentioning this: "... through a stainless-steel tubing inlet system **equipped with a PM2.5 size-cut cyclone (URG-2000-30ED)** and a Nafion dryer, sampling at 4.5-meter height with flow rate of 2 L min$^{-1}$. An **intermediate pressure lens (IPL) is utilized as aerodynamic lens allowing transmission of particles in the PM$_{2.5}$ fraction**…"

R1-1.2) What was the flow rate throughout the system? Was there an external pull for the ACSM to reduce residence times? What type of tubing was used? Was the inlet heated or not? Any concern about temperature gradients between inlet outside and instrument inside?

The average flow rate in the stainless-steel tubing inlet system and in the sample line are 2 L min$^{-1}$ and 1.22 cm$^3$ s$^{-1}$ respectively. We have added text to line 123-124 and 130-131 of the paper mentioning this:

Line 123-124: "...instrument through a **stainless-steel tubing inlet system** equipped with a PM2.5 size-cut cyclone (URG-2000-30ED) and a Nafion dryer, sampling at 4.5-meter height **with flow rate of 2 L min$^{-1}$.**"

Line 130-131: "**The average flow rate in the sample line of the instrument is 1.22 cm$^3$ s$^{-1}$**"

There is no external pull or heating applied to the inlet system. The length of the inlet line between the roof to the instrument is approximately 3 m.

R1-1.3) Were the SMPS and ACSM on same or different inlets? If different, how far apart were the two inlets?

The instruments were on separate inlets, which were mounted at similar heights (4.5 m) and adjacent to each other (maximum 3 m in lateral distance). We have added text to the paper mentioning this:

Line 152-153: "... **at 4.5-meter height** sampling with a flow rate of 16.7 L min$^{-1}$. The SMPS inlet is placed **approximately 3 m in lateral distance from the ACSM instrument inlet.**"

R1-1.4) Was a drier used for either or both instruments?

Both inlet systems use stainless steel tubing and are equipped with a Nafion dryer. The difference that may impact the sampling is the use of the PM$_{2.5}$ cyclone and lower flow rate (2 L min$^{-1}$) in the ACSM inlet, which is not the case for SMPS inlet that used PM$_{10}$ cyclone and higher flow rate. Details are described in the revised paper:

Line 124 (ACSM): "... and **a Nafion dryer**, sampling at 4.5-meter height…"

Line 152 (SMPS): "... cyclone and **Nafion dryer** at 4.5-meter height…"

R1-1.5) More information needs to be included about the SMPS, as it is one of the key instruments. This includes type of DMA and CPC, resolution, type of column, software for analysis, type of neutralizer source, etc.

This information has been added to the revised paper through Section 2.2.2., line 151-155:

"Ambient air was sampled using a stainless-steel inlet equipped with $PM_{10}$ size-cut cyclone and Nafion dryer at 4.5-meter height sampling with a flow rate of 16.7 L $min^{-1}$. The SMPS inlet was placed approximately 3 m lateral distance from the ACSM instrument inlet. The instrument consists of **a Vienna-type differential mobility analyzer (DMA) and a butanol-based TSI condensation particle counter (CPC) 3750.** The flow rate in the instrument is 1.0 L $min^{-1}$. The TSI CPC 3750 has the collection efficiency of 100% at the first selected and reported size of 10 nm.

The raw dataset was processed using a linear multiple charge inversion algorithm to derive the particle number size distribution (PNSD or $dN/dlog(D_p)$) (Wiedensohler et al., 2012; Pfeifer et al., 2014). **The MPSS inversion algorithm version 2.1**3 was utilized to obtain final PNSD from the raw dataset. The final PNSD has 5-minute time resolution and covers 71 geometric mean diameters ($D_p$) from 8 nm to 853 nm. The particle number concentrations (dN) for individual $D_p$ were then calculated by multiplying PNSD with $dlog(D_p)$ values for each $D_p$."

R1-1.6) What are the limits of detection for everything? E.g., were any of the measurements at or below LOD for when trying to investigate NPF?

According to Fröhlich et al., 2013 and ToF-ACSM user guide, the detection limits for aerosol species can be performed by putting a high-quality filter on the sample inlet and acquiring data for several hours. The detection limit is three times the standard deviation of the "zero" signal. However, we have not yet performed such an LOD study for our ACSM. With similar instrument configuration with Zheng et al., 2020 and Fröhlich et al., 2013, we assume that the detection limits are similar to their results. Details are described in line 142-143 in the revised paper: **"... where the values are 198–351.8 ng $m^{-3}$ for Org, 182–470.3 ng $m^{-3}$ for $NH_4$, 21–41.8 for $NO_3$, 18–33.6 ng $m^{-3}$ for $SO_4$, and 11–31.4 ng $m^{-3}$ for Cl."**

There is no LOD study either for the SMPS employed in this study. The TSI CPC 3750 can detect particle sizes as small as 7 nm ($D_{50}$), while the SMPS system can detect particle sizes in the range between 10 nm to 800 nm with practically 100% counting efficiency. Details are described in line 155 in the revised paper: **"The TSI CPC 3750 has the collection efficiency of 100% at the first selected and reported size of 10 nm."**

As the site is situated in the central Netherlands where there is a high aerosol concentration, we expect most measurements significantly exceed the LOD.

R1-2) Some further discussion or details also needs to be included for PMF. This includes:

R1-2.1) Why were the total mass concentration for the inorganics instead of their ions used?

Our main intention in this study is to look at bulk species and size distribution, but then the organic spectrum is also included to learn more about the variation in the organic composition. We found it simplest therefore to use the Tofware provided inorganic species alongside the organic spectrum.

R1-2.2) Why was potassium used? It is generally related with surface ionization of the vaporizer. Evidence that the potassium was from aerosols and not surface ionization should be included in the SI.

We were enthusiastic seeing the possibility of including potassium concentration in our PMF analysis, in hopes that it can infer any possibility of biomass burning evidence in the factors. However, we cannot prove that the potassium signal was coming from aerosols and not surface ionization. No RIE study was conducted for potassium. After considerations from the referees' comments and our further literature research, we decided to remove potassium from the PMF input matrix and not use it in this analysis after all. The removal of potassium as variable does not affect the main results. Therefore, "**S4. Correlation of chloride-sulfate and potassium-organic species"** section is removed from the SI.

R1-2.3) Why were the 18 bins selected for the SMPS? Was this due to the software, or did this provide the optimal data for analysis? Would fewer bins be better or worse? Further, some more discussion about the weighting of the SMPS data and errors in the SI would be beneficial.

The reason for the selection is mainly because the authors want more bins in smaller particle sizes. There are 71 mean geometric particle diameters ($D_p$), and 18 bins seemed to be the right number to still have good resolution in smaller sizes and not have too many bins.

Prior to analysis, the values and errors of species mass concentrations and particle number concentrations in the matrix ($X_{ij,input}$) were downweighted by dividing the dataset with a downweighting constant (DWC) to get the final values and errors ($X_{ij,DW}$),

$$X_{ij,DW} = X_{ij,input}/DWC$$

Previously, the DWC were chosen arbitrarily, purely by trial and error. Several DC values have been tested from 10-100 for species mass concentrations and 5000-100000 for particle size distributions. In the submitted results, the value of DWC = 10 for species mass concentration and DWC = 50000 for particle size distribution were the chosen as they give reasonable PMF solutions. However, despite the reasonable PMF solution, in most cases the previous solutions were not convergent with any $Q/Q_{exp}$ values.

The goal of downweighting is to normalize the magnitude of newly introduced variables in comparison to the organic mass spectrum, by decreasing their magnitude in reference to the spectrum. The dataset from September 2021 is used for the calculation as it represents the lowest concentration of all chemical species among analyzed periods (see Table (1) below). In addition, the 95% percentile concentration is selected for the calculation to avoid including outliers in the dataset. The concentration of m/z 44 fragment ($C_{f44}$) is chosen to represent the organic mass spectrum dataset for the calculation of DWC as it generally has the highest average peak among organic fragments. For the inorganic mass concentration, nitrate concentration ($C_{NO3}$) is selected as the reference for the inorganic chemical species since it

generally has the highest concentration among measured inorganics. The PNSD in the 51-65 nm size bin ($C_{p51-65}$) is chosen for the particle size distribution, also for the same reason.

The PNSD has a different unit compared to other variables (particles cm$^{-3}$ instead of µg m$^{-3}$). However, we disregard the unit as we are only interested in seeing the particle size concentration variation over the course of the time and how the sizes are being distributed in the profiles in the PMF solution. For downweighting, the DWC for PNSD variables is multiplied by 100 (so DWC = 83527.07) to tune the value so that it would have similar magnitude to the DWC value from the trial and error (DWC = 50000) and assuring convergent PMF solutions.

Table (1). The determination of downweighting constant (DWC) for inorganic species mass species and particle number size distribution (PNSD) in the PMF input matrix. The values are obtained from September 2021 dataset and applied for all analyzed periods.

| Variable | Value |
|---|---|
| $C_{f44;95\%}$ (µg m$^{-3}$) | 1.43 |
| $C_{NO_3;95\%}$ (µg m$^{-3}$) | 4.95 |
| $C_{p51-65;95\%}$ (cm$^{-3}$) | 1198.34 |
| $DWC\ inorganic = \dfrac{C_{NO_3;95\%}}{C_{f44;95\%}}$ | 3.45 |
| $DWC\ PNSD = 100 \times \dfrac{C_{p51-65;95\%}}{C_{f44;95\%}}$ | 83527.07 |

The detailed description of PMF variables downweighting is covered in the revised document, section **S3 "PMF variables downweighting"**, line 129-160 SI.

R1-3) Looking at the PMF organic profiles for all seasons, many of the profiles look very similar and/or like they are split solutions. E.g., in Fig. 3, the mass spectra for organics for solution F6, F4, F2, and F1 look nearly identical, and there are mass spectra for organics that look nearly identical in the SI. Should these solutions be combined? Also, the profiles are generally surprising looking as they do not look like profiles expected for ambient aerosol. Since there can be a potential bias in the CO2+ signal from CV, was m/z 44 (and associated ions) downweighted the similar amount as is typical for SV or downweighted more? Inclusion of the time series of each profile and SMPS data associated with that factor in the SI would also be extremely beneficial here in further evaluating and understanding if each solution is unique and real.

As our main intention of this study is to separate aerosol composition based on their particle size distribution, the priority criteria to choose the PMF solution and factors are based on the particle size, not the organic spectrum. Factors with similar organic mass spectrum should not be combined because they have different size distribution and composition. They just appear to have similar/same organic spectra but in different sizes. Details about the determination of PMF solution can be found in the revised SI, section **S3 "PMF variables downweighting"**, line 129-160 SI.

We are also surprised to see this result and that is the reason why we chose to only look at m/z 44 and 43 to infer the factor oxidation level. However, we are still interested in inferring the organic spectrum profile from each factor profile while still having the size distribution.

Based on the suggestion from the referees regarding: 1) the organic spectrum profiles that are not expected for typical ambient aerosol, 2) factors that look identical to each other, 3) lack of $Q/Q_{exp}$ values, and 4) consideration from the authors, we decided to redo the analysis with these details,

1. **Organic spectrum (m/z 12 to 120).**

   The organic spectrum will represent the profile of organics and their mass. In the submitted results, we only use m/z from 12 to 100 in PMF analysis, despite UMR-ToF-ACSM providing a mass spectrum with m/z up to 200. From additional literature reading (Xu et al., 2019; Zheng et al., 2020; Li et al., 2023), many only limit their m/z to 100 and 120, and thus we decided that we extend our m/z to 120 to be able to compare our results mainly with Zheng et al., 2020 and Joo et al., 2021. All m/z's larger than 120 are still excluded in PMF analysis due to low contributions to OA mass especially during summertime (our analysis does not include any winter months and therefore BBOA emission is not expected).

2. **ACSM inorganic species concentration ($NH_4$, $NO_3$, $SO_4$, Cl).**

   After several reruns to update our results, we noticed that by taking out the "Org" variable from the PMF input matrix, the PMF resolves better factors, and more solutions are converged. Having both organic spectrum and "Org" mass concentration means we are having a double contribution of Org to the PMF solution. We hypothesize that the introduction of "Org" mass concentration may be the origin of identical organic mass spectra. We concluded that we should remove the "Org" variable for more correct fit.

   We also removed potassium for the reason explained in R1-2.2.

3. **The particle size distribution.**

   We continued using the 18 size bins option for the particle size distribution. There are some updates to the dataset where we noticed that the last two PNSD size bins were miscalculated and the magnitude of the PNSD were higher. The description of particle size distribution concentration errors in the PMF input matrix is also changed from "population standard deviation" into "standard deviation". However, these circumstances do not have a huge impact on the results.

The new results suggest that the best PMF solution was found to have 4 factors for May 2021 (Fig. 2, line 243-253), June 2021 (Figure S1, line 6-14 SI), and September 2021 (Figure S2, line 15-23 SI). We mainly take into account size and inorganic composition, but also the organic profile interpretation beyond comparing f($CO_2^+$) among factors. By mainly comparing our organic mass spectrum results with Zheng et al., 2020 and Joo et al., 2021, the new 4-factor solution shows that the smallest size-driven factor (F4) possess LO-OOA profile while the larger size-driven factor (F3) possess HOA profile. The bulk/composition-driven factors F2 has LO-OOA profile while F1 has MO-OOA profile.

We did not perform the downweighting in the previous version of the paper as we intended the results to be fully unconstrained without any modification. In the new PMF runs, we perform the default downweighting for m/z 44, 28, 18, 17, and 16 signal in UMR-AMS provided by PETv3.08 The details are covered in line 157-160 SI: **"The downweighting procedure is applied for m/z 44, 28, 18, 17, and 16 signals in the organic mass spectrum** as provided by PETv3.08 during PMF input matrix preparation (Ulbrich et al., 2009). **A correction calculation for capture vaporizer (CV) is also opted at the end of the PMF analysis** to account the additional thermal decomposition in smaller fragments." Bootstrap run results are incorporated with the complete 4-factor solution in May 2021 (Fig. 2, line 243-253), June 2021 (Figure S1, line 6-14 SI), and September 2021 (Figure S2, line 15-23 SI).

The time series of each profile and PNSD of one of the size bins appearing in the factor to show the correlation between the factor and the size are shown in **Figure S5 in SI, line 35-39 SI.**

R1-4) One very large concern is what aerosol diameters are being observed with the ACSM. If it is using a PM1 lens, any aerosol below 40 nm is not observed, and aerosol between 40 and 70 to 100 nm is only fractionally observed (e.g., approximate linear growth in the amount of aerosol observed with diameter to 70 - 100 nm). However, if it is PM2.5 lens, the ACSM will only observe 100% of aerosol for diameter > 110 nm. Thus, any aerosol observed for most of the solutions/modes for NPF are very surprising. Since many of the figures show potential contribution of "large" particles (>100 nm diameter) showing small contribution to the factor, how much volume is the small, large particle, contribution adding?

E.g., is the volume large enough that that is what is leading to the aerosol being observed by the ACSM?

According to a study conducted using a ToF-ACSM in the same configuration by Xu et al., 2017, the PM$_{2.5}$ lens in the ToF-ACSM transmits particles with vacuum aerodynamic diameter (D$_{va}$) between 100 nm and 3 μm with efficiency above 50%. Meanwhile, the SMPS instrument samples mainly particles with diameters ranging from 10 to 800 nm (counting efficiency 100%), a different range compared to PM$_{2.5}$ ToF-ACSM. This, and the fact that smallest particle size bins contribute significantly less volume and therefore mass, means that the smaller particles counted by SMPS are represented less in the detected aerosol composition by ACSM (see Fig. (1) for illustration).

[Figure]

Figure (1). (a-b) Normalized average particle size distribution of two size-driven factors of F4 (maroon) and F3 (turquoise) across periods. The line plot shows particle number concentration (dN) fraction on each size bin. The thick histogram represents cumulative particle number concentration (dN) fraction of as the particle size increases, while the thin histogram represents cumulative particle volume (dN×(4/3)×π×(Dp/2)$^3$) fraction of each size-bin median diameter. The vertical dashed dark grey line divides the particle diameters where particles are transmitted with <50% efficiency on the left (diameter less than ~100 nm) and with at least 50% efficiency by PM$_{2.5}$ lens of ToF-ACSM on the right (diameter more than ~100 nm). (c-h) Pie charts showing mass percentage of each aerosol species contributing to each size-driven factor in May 2021, June 2021, and September 2021. Green represents organics (Org), orange represents ammonium (NH$_4$), dark blue represents nitrate (NO$_3$), dark red represents sulfate (SO$_4$), and pink represents chloride (Cl). F4 are dominated by ammonium sulfate while F3 are dominated by ammonium nitrate. The mean PMF fractions and their standard deviations are shown indicating the mean contribution of the factor to the total reconstructed PMF mass.

The small-sized particles will make a negligible contribution to the PM$_{2.5}$ mass and the larger particles will always dominate the particle volume size distribution, regardless of whether the finer particles are efficiently sampled or not by ACSM. Nevertheless, although the ACSM does not directly measure the finest particle composition, the factor still illustrates the bulk chemical composition that occurs during and favors the formation and growth of new particles.

We addressed this issue in whole details, including the figure illustrating the particle size distribution in the concerned size-driven factors in the revised paper, **Section 3.3.1, starting from line 346 to 363: "Some concerns may arise due to the fact that the ACSM and SMPS measure different particle size ranges, especially the smaller sizes which are the major interest of this study…"**

R1-5) Looking at the progession of the NPF with the ACSM data is very surprising and needs further discussions. Some specific questions that need to be addressed are listed below:

R1-5.1) How does the composition shift entirely from sulfate to organics or sulfate/organics to nitrate between F6 to F5? What happened to the sulfate? Looking at the solutions, it appears that F6 --> F4 and may be F5 --> F3; however, as it is presented and discussed, it appears the NPF event goes from F6 --> F5 --> F4 --> F3.

R1-5.2) Similarly, what happened to f(CO2)? Highly oxyenated material may be necessary for

the initiation of NPF; however, it should not completely disappear as compounds with higher volatility, lower f(CO2) condenses onto the aeorosol.

We thank the reviewer's comments in 5.1. and 5.2. The response below addresses the comments. We propose that there are several ways to explain the pathway of new particle formation and growth as illustrated by Figure (2) below.

[Figure]

Figure (2). Proposed new particle formation and growth pathway, either sequential (red arrows) or simultaneous (dark grey arrows) formation of F4 and F3.

[Figure]

Figure (3). Selected timeseries windows during which new particle formation (NPF) events were detected by the scanning mobility particle sizer (SMPS) resembling 'banana' shapes in May 2021.

1. **Sequential pathway** (Fig. (3)a-c):
   The high occurrence of ammonium sulfate and oxidized organic molecules in the aerosol phase can be observed during nucleation-mode F4 episodes, marking the beginning of NPF. This is related to ammonium sulfate formation from $NH_3$ and $H_2SO_4$, and uptake of

oxidized organic compounds. We can consider F3 as a "sequential" pathway of F4 growing in size; this sequential nature is observed in some NPF events shown in Fig. S4, when F3 peaks after F4. F4 grows into F3 when nitric acid and/or organic nitrates and hydrocarbon-like semi-volatile organic compounds is dominant in the aerosol composition.

2. **Parallel pathway**
   Another explanation is to consider F3 emerging directly from ammonium nitrate as a "parallel" nucleation pathway. Other studies have observed this nucleation mode to occur very rarely and only in the free troposphere, at lower temperature and very clean air conditions, through reaction between nitric acid and $NH_3$ (Höpfner et al., 2019; Wang et al., 2020).

3. **Combined pathway** (Fig. (3)d-f), where F3 emerges from F4, but the aerosol phase rapidly favor the pathway of growing by uniquely through ammonium nitrate condensation.There have been report in chamber experiments and theoretical study supporting this interpretation where NPF occurs with small involvement of sulfate despite the presence of $SO_x$ or $H_2SO_4$ (Liu et al., 2018; Wang et al., 2020, 2022).

In terms of organic compounds, the oxygenated organic compounds responsible for new particle formation do not disappear but gets "diluted" by the condensation of compounds with lower $f(CO_2)$ (in this case saturated organic compounds making up HOA profile in F3 on the new results), and therefore it may appear to be lower (or gone) on the factor with larger particle size range. As the aerosol age into composition-driven factors (F2 and F1), the oxidation level increases again from HOA (F3) to SIA+LO-OOA (F2) and/or MO-OOA (F1) in the bulk composition.

The whole section is covered in **"3.3.4 NPF and growth pathway", line 424-442** in the revised paper.

R1-6) Looking at the time series of SMPS number concentration vs time (Fig 5 & SI), it is not clear what has lead to some events being specifically selected as NPF and other times where there is what appears to be rapid particle formation not being selected as an NPF. For example, in Fig. 5, why was the third event selected as it appears it only went to 30-40 nms and stopped but later times (after 5/30) not selected?

R1-7) Fig 6 and associated figures in SI, it is surprising how the normalized mass spans what appears to be a larger time frame than the NPF event. E.g., Fig. S4 shows that the events are ~ 4 - 6 hours; however, looking at Fig. 6 (and associated figures), it seems that it takes a full 12 hrs to go from F6 --> F3. Clairification in how this figure/results are related to NPF needs to be further detailed.

We thank the referee for the comments 6 and 7. In the original manuscript, we indicated several NPF events with yellow squares to demonstrate that the PMF solution matches the PNSD time series. However, we do not point out all of them to avoid crowded figures. To avoid misunderstanding, we remove entirely the rectangles indicating NPF events (see Figure 4, line

333-337, and Figure S3, line 24-28 SI), since some examples of zoomed time window in NPF events are covered in Figure S4, line 29-34 SI.

The relation between NPF events length in the timeseries (Fig. S4, line 29-34 SI) and the diurnal cycle of size-driven factors (see Fig. 5g-l, line 399-404) of the new results are addressed in line 431-433 in the revised paper: "...reveal that particle formation and growth takes around **6 to 12 hours to complete** (see Fig. S4, line 29-34 SI)"

R1-8) Section S2. Further clarification needs to be added in this section to discuss the thermodynamics vs kinetics that may be controlling NPF and the aerosol composition in general. It is recommended that Weber et al. (2016) and Pye et al. (2020) are reviewed and incoporated in the discussions here, for the following reasons.

We thank the reviewers for this input. We have added this information to the section S2 under subsection **"Ammonium sulfate and nitrate aerosol formation"**, line 102-126 SI.

R1-8.1) Are the values 0.99 and 0.98 statistically different, considering the overall uncertainties with the ACSM?

R1-8.2) It is nearly impossible to say anything about aerosol acidity in the boundary layer just with charge balance calculated with the ACSM/AMS. E.g., it was not until the ammonium balance dropped below 0.65 could aerosol acidity be directly related to the charge balance measured on the AMS (ACSM) (Schueneman et al., 2021).

R1-8.3) Though NO3 and SO4 would be with other cations, generally, both the cations and anions would be not easily observable due to the higher boiling point and the aerosol being more refractory. It would be recommended to say that both the cations and anions from these salts would be slowly detected and not "not detected." (Line 939 SI).

This response answers comments 8.1-8.3. We thank the reviewer for the input for the ion balance ratio/ammonium balance values interpretation in our article. There are indeed large enough uncertainties in theACSM measurements that we can consider the ammonium balances from the three periods close to unity. We therefore decided to interpret that the bulk ion charge is fully neutralized in the three periods. For inputs on comments 8.2 and 8.3, we decided to include this information in the revised text:

Line 85-88 SI: "It also may be caused by excess aerosol acidity (Farmer et al., 2010; Docherty et al., 2011) although **the clear relationship between NH4_bal and high aerosol acidity (pH < 0) is observed for mass spectrometry measurements only when NH4_bal < 0.65 (Schueneman et al., 2021).**"

Line 83-85 SI: "The anions **can form refractory compounds with other cations (e.g., NaNO$_3$, Na$_2$SO$_4$, Ca(NO$_3$)$_2$) and thus be less efficiently detected by the spectrometer**, or exist in form of ..."

R1-8.4) Line 945 - 950. This needs to be rephrased as both the association of sulfate with a base is both kinetically and thermodynamically controlled (see Weber et al., 2016 and Pye et al., 2020). Sulfuric acid will first react with a base (either ammonia or an amine) very quickly; then, it will more slowly form the ammonium sulfate or double-amine sulfate. E.g., > 100 ug m^-3 NH3 was estimated to be needed to make pure ammonium sulfate. Instead, it will be a combination of ammonium sulfate and bisulfate in the aerosol phase. Further, a combination of factors (temperature, relative humidity, ammonia, and ammonium) will play in the role to start having ammonium nitrate in the aerosol phase, which is best explained with a thermodynamic model. Even at "low" pH (~2), ammonium nitrate will be present even though the sulfate is not pure ammonium sulfate. Thus, it is not as straightforward that all the ammonia reacts with sulfate to form ammonium sulfate and the remainder then reacts with nitrate.

We have added a subsection in SI explaining the formation of ammonium sulfate and nitrate, starting from line 102 to 126 SI: "The ammonium sulfate and nitrate aerosol formation can be explained...**The formation of ammonium sulfate and nitrate aerosols involves the buffering capacity of semi-volatile NH$_3$ partitioning between the gas and particle phase, reacting with H$_2$SO$_4$ and HNO$_3$ forming aerosols…**"

R1-8.5) The terms "nitrate excess" and "sulfate-rich" also are hard to follow for the reasons discussed in 8.4.

Since the ion balance ratios are considered to be unity and the sulfate-to-nitrate ratio is out of use, the terms are not used anymore in the revised paper.

R1-9) What does an "orgnaic-rich" period mean, in that it was related to ammonium? Why was ammonium used to normalize and determine organic rich vs poor? Clarification in what this chemically means should be addressed.

The term "organic-rich" is coined to purely describe that,relative to other seasons, summer has much higher OA concentration compared to IA. One of the ways to compare it is to look at the ratio between Org and NH$_4$ (m Org/m NH$_4$), since IA in the Netherlands are mainly composed of ammonium. This term has no relation with chemistry. To avoid misunderstanding, we decided to modify the value to be the ratio between Org and all IA species detected by ACSM (m OA/m IA). Among the three periods analyzed, June (summer) is observed to have the highest ratio, meaning it is organic-rich in respect to other seasons.

Minor

1) Line 333, believe September should be fall instead of summer?

We thank the reviewer for pointing out the mistake. The sentence is corrected and now positioned in line 315-316: "In summer (June), F4 accounts in average 14.9% of total reconstructed PMF mass while in spring (May) and **autumn (September)**, they only represent 11.8% and 7.8%, respectively."

2) line 364, what is quiet NPF?

Removed, to avoid confusion, as it was not necessary for the explanation.

3) Sect 3.3 Title should be F6, F5, F4, F3 and not F7, F6, F5, F4

Updated with new PMF solutions, and sect 3.3 is now titled "**3.3. Size-driven factors (F4 and F3)" (line 299).**

4) Line 225. A discussion about what happened to the sulfate and why it is suddenly poor in Sept should be included

We decided to not use sulfate-rich and sulfate-poor terms anymore since we agree with the reviewer that the terms are confusing. With the new PMF results, we also find that it is unnecessary to incorporate it in the article.

However, we would like to reply to this discussion. The best explanation for the decrease in sulfate in September (hence previously described as sulfate-poor) is due to the increase of atmospheric $NO_x$ concentration. Referring to Table S1 in line 93-100 SI, the molar ratio between free atmospheric $NH_3$ and $SO_2$ concentrations are pretty constant throughout the seasons (~1.2 x $10^{-2}$ to 1.5 x $10^{-2}$).

However, in September, while a decrease of $NH_3$ and $SO_2$ concentrations were observed, the $NO_x$ concentration increases. It would mean that there is more nitric acid and less sulfuric acid available in comparison to previous seasons, and therefore higher nitrate aerosol concentration is observed. Since the original manuscript used the sulfate-to-nitrate ratio, it made it appear that there was a "decrease" of $SO_2$ and $SO_4$ (hence sulfate-poor), where actually it was due to both a decrease in sulfate and an increase of nitrate.

References:

Fröhlich, R. et al.: The ToF-ACSM: a portable aerosol chemical speciation monitor with TOFMS detection, Atmos. Meas. Tech., 6, 3225–3241, https://doi.org/10.5194/amt-6-3225-2013, 2013.

Höpfner, M. et al.: Ammonium nitrate particles formed in upper troposphere from ground ammonia sources during Asian monsoons, Nat. Geosci., 12, 608–612, https://doi.org/10.1038/s41561-019-0385-8, 2019.

Li, Z. et al. Insights into the compositional differences of PM1 and PM2. 5 from aerosol mass spectrometer measurements in Beijing, China. Atmospheric Environment, 301, 119709, 2023.

Liu, L., et al.: The role of nitric acid in atmospheric new particle formation, Phys. Chem. Chem. Phys., 20, 17406–17414, https://doi.org/10.1039/C8CP02719F, 2018.

Pye et al. The Acidity of Atmospheric Particles and Clouds. Atmos. Chem. Phys. 20, 4809 - 4888. doi:10.5194/acp-20-4809-2020, 2020.

Schueneman et al. Aerosol pH Indicator and Organosulfate Detectability from Aerosol Mass Spectrometry Measurements. Atmos. Meas. Tech. 4, 2237 - 2260. doi:10.5194/amt-14-2237-2021, 2021.

Ulbrich, I. M. et al.: Interpretation of organic components from Positive Matrix Factorization of aerosol mass spectrometric data, Atmos. Chem. Phys., 9, 2891–2918, https://doi.org/10.5194/acp-9-2891-2009, 2009.

Wang, M. et al.: Rapid growth of new atmospheric particles by nitric acid and ammonia condensation, Nature, 581, 184–189, https://doi.org/10.1038/s41586-020-2270-4, 2020.

Wang, M. et al.: Synergistic HNO3–H2SO4–NH3 upper tropospheric particle formation, Nature, 605, 483–489, https://doi.org/10.1038/s41586-022-04605-4, 2022.

Weber et al. High Aerosol Acidity Despite Declining Atmospheric Sulfate Concentrations Over the Past 15 Years. Nature Geosci. 9, 282-285. doi:10.1038/ngeo2665

Xu, W. et al.: Laboratory characterization of an aerosol chemical speciation monitor with PM 2.5 measurement capability, Aerosol Science and Technology, 51, 69–83, https://doi.org/10.1080/02786826.2016.1241859, 2017.

Xu, W. et al. Changes in aerosol chemistry from 2014 to 2016 in winter in Beijing: Insights from high‑resolution aerosol mass spectrometry. Journal of Geophysical Research: Atmospheres, 124(2), 1132-1147, 2019.

Zheng, Y. et al.: Characterization of anthropogenic organic aerosols by TOF-ACSM with the new capture vaporizer, Atmos. Meas. Tech., 13, 2457–2472, https://doi.org/10.5194/amt-13-2457-2020, 2020.

—--------

**Reviewer #2**
Nursanto et al. provided insights in new particle formation of different chemical species by combining organic aerosol mass spectrum, inorganic mass concentration from ACSM, and 18 particle size bins from SMPS. It suggests that the small-size particles are related to the transport of SOx, NH3, and some organic precursors. Moreover, nitrate plays an important role while particle size grows. However, there are still some fundamental questions that need to be addressed to draw such conclusions.

General comments:

R2-1: PM2.5 inlet of ACSM is subject to a significant loss for particles that have a small size, do you believe SMPS and ACSM are measuring the same thing? Do you believe these small particles in F6 were actually measured by ACSM?

See reply R1-4 above.

R2-2: A more detailed description of how to balance the estimated error from different instruments is required (in this case, since mass conc. of inorganic were used, it's like combining three different datasets).

See reply R1-2.3 and R1-3 above.

R2-3: A more detailed description of the number of factor decisions is required.

We run the analysis several times and the optimum number of factors is decided based on the fact that it gives the most distinct results that are not redundant (repeated several times), yet still resolving particle based on size. The last two factors that are not size-driven always appear regardless of factor number, and they are always bulk composition (F2) and bulk organic aerosol (F1). The details are covered in line 163-168 SI in the revised paper:

"**The optimum p is selected first based on the lowest residuals and local minima** ($Q/Q_{exp}$) of the PMF solutions. With the introduction of inorganic species mass concentration and particle number size distribution (PNSD) variables, we suggest that **the PMF solution also must include at least two factors that show a significant signal of particle size distribution**, or size-driven factors. A minimum of two size-driven is required in order to study new particle formation and growth. Lastly, **the lowest $Q/Q_{exp}$ should not be accepted if it contains redundant factors with very similar organic and inorganic profile**."

R2-4: Why only up to m/z 100 were used? ToF-ACSM has data up to m/z 200 that potentially can increase the capability of better PMF factor separations.

See reply R1-3, with the heading "**1. Organic spectrum (m/z 12 to 120)".**

R2-5: The PMF solutions, especially the OA factors are not convincing, even with CV-ACSM, literature has shown successful PMF analyses with reasonable solutions in both China and Atalanta. The PMF factors are not well-separated and seems like the authors define the factors heavily based on the SMPS data. Authors need to show that factors are not mixed from time series, diurnal, and mass spectrums. Currently, the mass spectrum from OA suggests they are mixed. In addition, bootstrap should be conducted to demonstrate that current results are robust and stable.

We thank the reviewer for the comment regarding the factor's organic profile. We performed new PMF runs as described in R1-3 and have chosen a 4-factor solution that weighs size and inorganic composition, but also taken into account the organic profile interpretation beyond comparing f($CO_2^+$) among factors. By mainly comparing our organic mass spectrum results with Zheng et al., 2020 and Joo et al., 2021, the new 4-factor solution shows that the smallest size-driven factor (F4) possess LO-OOA profile while the larger size-driven factor (F3) possess HOA profile. The bulk/composition-driven factors F2 has LO-OOA profile while F1 has MO-OOA profile.

Bootstrap run results are incorporated with the complete 4-factor solution in May 2021 (Fig. 2, line 243-253), June 2021 (Figure S1, line 6-14 SI), and September 2021 (Figure S2, line 15-23 SI).

R2-6: How do PMF results look like when you only use the OA mass spectrum? Does it also provide a 6-factor solution that supports your current conclusions (e.g., K related to biomass burning)

During the revision process, we have decided to remove potassium from our input matrix (see response R1-2.2 above) and therefore there is no interpretation of K relating to biomass burning.

Line 101: Would be great if you can provide the average temperature of May when you say it was characterized as moderate spring temperatures in the text. Same for the highest temperature for June and the warm temperature for Sep.

Accepted and added in line 103-107: "In general, May 2021 was characterized by moderate spring temperatures **(11.8 °C on average)** with scattered precipitation transitioning into the warmer summer period. June 2021 had the highest temperatures **(18.7 °C on average)** and was the sunniest of the three periods, reflecting summer weather. September 2021 showed warm temperatures **(16.2 °C on average),** with less radiation and precipitation compared to May 2021."

Line 124: These citations are rather for collection efficiency correction based on SV. I feel like it's better if you can explain the CE in your word instead of citing this literature since you are not using their methods to apply CE correction.

Accepted and rephrased the paragraph in line 125-128:

"**The instrument uses capture vaporizer (CV) to increase the particle collection efficiency (CE) compared to standard vaporizer (SV) (Jayne and Worsnop, 2016). By having a narrow entrance, the CV increases the particle collision events and thus increases the contact with the hot vaporizer surface, minimizing particles that bounce without evaporation (Hu et al., 2017) resulting in higher CE.**"

Line 129: Please cite James Allen's paper for the fragmentation table you used.

Accepted as shown in line 135-136, thank you.

Line 135-136: "The fractions of measured UMR signals were assigned to individual aerosol species using the fragmentation table **(Allan et al., 2004)**."

Line 133: How confident are you about your potassium signals from ToF-ACSM, I've barely seen any of the other studies report it. Did you also conduct RIE calibration for it?

See reply R1-2.2.

Line 146: mass-to-charge ratio?

Revised in line 163: "The 10-minute average matrices of UMR organic fragment mass spectra with **mass-to-charge ratio (m/z)** 12 to 120 were…"

Line 153: It's great that the authors considered balancing the variables from the different instruments, but the detailed description and how well the weighting should be discussed in this study. Because it is the key to ensuring the quality of the results.

See reply R1-2.3.

Line 161: As the most subjective part of the PMF, it would need more detailed descriptions and illustrations to justify your selection of the number of factors. Also, did you bootstrap your final solution to make sure your results are stable? This step is also important to make sure the solution is representative and robust.

See reply R2-3 for the determination of PMF solution description. Bootstrap run results are incorporated with the complete 4-factor solution in May 2021 (Fig. 2, line 243-253), June 2021 (Figure S1, line 6-14 SI), and September 2021 (Figure S2, line 15-23 SI).

Line 198-199: This statement of lower f44 and higher f43 is often referred to as HOA is simply false. There are lots of primary sources that could have this feature. Please rephrase.

We thank the reviewer's comment regarding the organic mass spectrum interpretation. We did a major revision in **Section 3.2.2. "Factor organic profiles"** that explains the primary organic aerosols (POA) and mentions some of the most common POA resolved from PMF analysis in line 282-285:

"**POA consists of various sources** which can be identified from the appearance… Some of the most common POA from PMF analysis are **hydrocarbon-like organic aerosols (HOA)**, …".

We then briefly mentions the similar characteristics of POA, and provide information of what can be used to distinguish different types of POA (including HOA) in line 285-292:

"**POA have similar characteristics of alkyl and alkenyl fragments** … **HOA as a type of POA is often correlated with anthropogenic combustion pollutants**, such as NOx and black carbon from vehicular emission…".

Line 200-203: There are quite a few studies of PMF using CV ACSM already, therefore, I think the authors cannot simply say that your results are not comparable with other works e.g., Joo et al., 2021 and Zheng et al., 2020. I'm still convinced that the CV ACSM should provide sufficient information to resolve reasonable PMF factors based on literature and some ongoing studies.

We thank the reviewer for pointing out these previous studies. We have detailed the response regarding comparing OA profile with other CV-ACSM studies (Zheng et al., 2020 and Joo et al., 2021) in reply R2-5.

Line 258-259: I have a hard time believing that OA correlates with K signal leads to biomass burning origin, not to mention how trustworthy the K signal is from the ACSM. If the K signal is so pronounced and you believe it is from biomass burning, you shall be able to resolve a biomass burning OA factor in Sep. I wonder if that's the case, otherwise, it is difficult to believe your statement.

See reply R1-2.2.

Figure 3, S1 and S2:

The diurnal plots for each factor shall be displayed side by side with the factor profiles for all three months to have a better comparison among factors to conclude F1 seems to be aged.

Accepted, thank you. Diurnal plots for each factor are now displayed next to the PMF factor profiles in Fig. 1

Figure 1:

Perhaps it's better to combine Fig 1 and 2 to have a better visualization of where the wind comes from.

Accepted, thank you. Fig. 1 and Fig. 2 now are combined as Fig. 1 (line 95-101).

References

Allan, J. D., Delia, A. E., Coe, H., Bower, K. N., Alfarra, M. R., Jimenez, J. L., Middlebrook, A. M., Drewnick, F., Onasch, T. B., Canagaratna, M. R., Jayne, J. T., and Worsnop, D. R.: A generalised method for the extraction of chemically resolved mass spectra from Aerodyne aerosol mass spectrometer data, Journal of Aerosol Science, 35, 909–922, https://doi.org/10.1016/j.jaerosci.2004.02.007, 2004.

Hu, W., Hu, M., Hu, W., Jimenez, J. L., Yuan, B., Chen, W., Wang, M., Wu, Y., Chen, C., Wang, Z., Peng, J., Zeng, L., and Shao, M.: Chemical composition, sources, and aging process of submicron aerosols in Beijing: Contrast between summer and winter, J. Geophys. Res. Atmos., 121, 1955–1977, https://doi.org/10.1002/2015JD024020, 2016.

Jayne, J. T. and Worsnop, D. R.: Particle Capture Device, 2016.

Joo, T., Chen, Y., Xu, W., Croteau, P., Canagaratna, M. R., Gao, D., Guo, H., Saavedra, G., Kim, S. S., Sun, Y., Weber, R., Jayne, J., and Ng, N. L.: Evaluation of a New Aerosol Chemical Speciation Monitor (ACSM) System at an Urban Site in Atlanta, GA: The Use of Capture

Vaporizer and PM 2.5 Inlet, ACS Earth Space Chem., 5, 2565–2576, https://doi.org/10.1021/acsearthspacechem.1c00173, 2021.

Zheng, Y., Cheng, X., Liao, K., Li, Y., Li, Y. J., Huang, R.-J., Hu, W., Liu, Y., Zhu, T., Chen, S., Zeng, L., Worsnop, D. R., and Chen, Q.: Characterization of anthropogenic organic aerosols by TOF-ACSM with the new capture vaporizer, Atmos. Meas. Tech., 13, 2457–2472, https://doi.org/10.5194/amt-13-2457-2020, 2020.

---

## Author Response (AR2)

**Second response to reviewers for "What chemical species are responsible for new particle formation and growth in the Netherlands? A hybrid positive matrix factorization (PMF) analysis using aerosol composition (ACSM) and size (SMPS)" by Nursanto, Farhan R; Meinen, R.; Holzinger, R.; Krol, Maarten C.; Liu, Xinya; Dusek, Uli; Fry, Juliane L. (Manuscript ID: EGUSPHERE-2023-554)**

The last modification results in the following changes in figure numbering:

| *Before revision* | ***After revision*** |
|---|---|
| Figure 1 and 2 | Stay as **Figure 1 and 2** |
|  | New figure entry as **Figure 3.**

Figure 3. Relation between the size-driven factors (F4 and F3, linked to NPF and growth) with the composition-driven factors (F2 and F1, linked to the bulk atmospheric aerosol composition) illustrating aerosol formation progress. |
| Figure 3-6 | Now **Figure 4-7** |

**Responses to comments:**

1) The authors interchange MO-OOA and LO-OOA for the F4 factor for NPF. Please check which one it is. Further, it would be good to explain, if it is LO-OOA, why it is less-oxidized, as it is typically thought that very low volatility compounds are needed NPF. As it is not straight-forward to say that LO-OOA means it is semi- or low-volatile, understand this can be complicated, but would be beneficial for the story to reduce confusion or provide motivation for future studies.

We thank you for the comments. We meant to refer to the F4 factor as LO-OOA and not MO-OOA, for all seasons. Confusion was produced because of a mistaken label in the Figure 6; we thank the reviewers for flagging this and have corrected it. We also noticed this error in the last paragraph of Sect. 3.2.2 mentioning that there are in total 1 LO-OOA and 2 MO-OOA factors instead of 2 LO-OOA and 1 MO-OOA factors. The corrected paragraph includes the line:

"The PMF analyses in this study resolved one POA factor (as HOA factor) and three SOA factors **(two LO-OOA factors and one MO-OOA factor)** across periods, …"

We also added some text explaining our interpretation of what the reviewer notes as a surprising presence of LO-OOA rather than MO-OOA in the nucleation mode factor, in the first paragraph of section 3.3:

"The organic mass spectrum profile from each size-driven factors and their diurnal cycles in each period are shown in Fig. 6. Across seasons, LO-OOA is part of bulk composition related to nucleation-mode particles. The factors are assigned as LO-OOA due to their $f_{44}/f_{43}$ values compared to other OOA factors (see the triangle plot in Fig. S6). The LO-OOA F4 profile resolved in this study is comparable to LO-OOA resolved in other aerosol mass spectrometry studies using CV (Zheng et al., 2020; Joo et al., 2021), although fragments with m/z > 50 are less prevalent. Several aerosol chamber experiments have reported that lower volatility and highly oxygenated organic molecules from biogenic and anthropogenic organic precursors play a dominant role in new particle formation and growth (Schobesberger et al., 2013; Ehn et al., 2014; Riccobono et al., 2014; Tröstl et al., 2016; Mohr et al., 2019; Pospisilova et al., 2020; Zhao et al., 2021). **In this study, however, we surprisingly observe LO-OOA rather than MO-OOA associated with nucleation. This could imply that organic compounds with less oxygenation are more abundant and condense on freshly nucleated particles in this region, or that the ToF-ACSM delineation between LO-OOA and MO-OOA does not directly correspond to volatility in this case.**"

2) The explanation about F4 and F3 starting at line 429 is extremely appreciated. I almost wish it was sooner, as seeing Fig. 3, Fig. 5, and the description about the factors prior to the explanation is occurring. I understand this may impact the flow of the paper, but not discussing why F3 does not looking like F4 with nitrate will cause confusion until the reader gets to line 429.

3) Comment 2) may be rectified by incorporating Fig (2) from the responses. I really appreciate the updated figures, including the Fig (1) from responses that is now Fig (3) in manuscript. Inclusion of Fig (2) provides a great summary and cartoon of the results and hypothesis the authors are discussing. Introduction of this Figure potentially before Fig (3) in the manuscript and a quick "overview" or something along that line could be of use for the readers.

Thank you for this suggestion; we now include the schematic figure in the manuscript, to illustrate how F4 and F3 could be connected in different pathways of particle nucleation and growth. We added this new Fig. 3 with caption at the end of section 3.2.1:
"

[Figure]

**Figure 3. Potential relationships between the size-driven factors (F4 and F3, linked to NPF and growth) with the composition-driven factors (F2 and F1, linked to the bulk atmospheric aerosol composition), illustrating multiple possible aerosol growth pathways. From F4 (mainly ammonium sulfate) and F3 (mainly ammonium nitrate), particles can grow into F2 (OA and IA mixed) and/or F1 (OA-dominant), either sequentially (dashed), in parallel (solid), or combined. The particle formation and growth occur through condensation of gaseous precursors or particle coagulation. An increase in organics and $NO_3$ in the bulk composition is observed as particles progress along these pathways."**

The accompanying text has been modified to refer to this figure at the end of Section 3.2.1.: "**We can summarize that the NPF and growth follow the pathway starting from F4 and F3 into F2 and F1 (bulk aerosol composition), likely through processes such as condensation of gaseous precursors (SOx, NH3, NOx, and VOCs and their reaction products) or particle coagulation (see Fig. 3). We note that this does not imply that all aerosol growth proceeds sequentially through these four factors; a more detailed discussion of possible NPF and growth pathways is found below in Sect. 3.3.4**"

And we have added reference to Fig. 3 in section 3.3.4.

**Additional minor changes:**

We in the meantime have conducted our own detection limit measurements, so we have updated this in the text section 2.2.1.:

**"The detection limits (measured similarly to Fröhlich et al., 2013) at 10-minute time resolution for this ToF-ACSM operating at Cabauw (a relatively polluted site in central Netherlands) are 0.38 µg m$^{-3}$ for Org, 0.12 µg m$^{-3}$ for NH$_4$, 0.07 µg m$^{-3}$ for NO$_3$, 0.11 µg m$^{-3}$ for SO$_4$, and 0.09 µg m$^{-3}$ for Cl."**

We have also updated the acknowledgements section and made other small edits throughout the manuscript; we have made no changes to the supplemental information but will re-upload the file.

---

## Author Response (AR3)

The submitted files are identical to the version reviewed, except for Fig. S10 in SI, which was corrected to fix a minor error in the percentages indicated in the pie charts of m, n, and p.